

# The climate of the Mediterranean basin during the Holocene from terrestrial and marine pollen records: A model/data comparison

**Odile Peyron[1], Nathalie Combourieu-Nebout[2], David Brayshaw[3], Simon Goring[4], Valérie Andrieu-Ponel[5], Stéphanie Desprat[6,7], Will Fletcher[8], Belinda Gambin[9], Chryssanthi Ioakim[10], Sébastien Joannin[1], Ulrich Kotthoff[11], Katerina Kouli[12], Vincent Montade[1], Jörg Pross[13], Laura Sadori[14], Michel Magny[15]**

[1] Institut des Sciences de l'Evolution (ISEM), Université de Montpellier, France

[2] UMR 7194 MNHN, Institut de Paléontologie Humaine 1, Paris, France

[3] University of Reading, Department of Meteorology, United Kingdom

[4] Department of Geography, Univ. of Wisconsin-Madison, Wisconsin, USA

[5] Institut Méditerranéen de Biodiversité et d'Ecologie marine et continentale (IMBE), Aix Marseille Université, Aix-en-Provence, France

[6] EPHE, PSL Research University, Laboratoire Paléoclimatologie et Paléoenvironnements Marins, Pessac, France

[7] Univ. Bordeaux, EPOC UMR 5805, Pessac, France

[8] Geography, School of Environment, Education and Development, University of Manchester, United Kingdom

[9] Institute of Earth Systems, University of Malta, Malta

[10] Institute of Geology and Mineral Exploration, Athens, Greece

[11] Center for Natural History and Institute of Geology, Hamburg University, Hamburg, Germany

[12] Department of Geology and Geoenvironment, National and Kapodistrian University of Athens, Greece

[13] Paleoenvironmental Dynamics Group, Institute of Earth Sciences, Heidelberg University, Germany

[14] Dipartimento di Biologia Ambientale, Università di Roma "La Sapienza", Roma, Italy

[15] UMR 6249 Chrono-Environnement, Université de Franche-Comté, Besançon, France

Correspondence to: O. Peyron (odile.peyron@univ-montp2.fr)



Abstract
Climate evolution of the Mediterranean region during the Holocene exhibits strong spatial and
temporal variability. The spatial differentiation and temporal variability, as evident from
different climate proxy datasets, has remained notoriously difficult for models to reproduce. In
light of this complexity, we examine the previously described evidence for (i) opposing
northern and southern precipitation regimes during the Holocene across the Mediterranean
basin, and (ii) an east-to-west precipitation gradient or dipole during the early Holocene, from
a wet eastern Mediterranean to dry western Mediterranean. Using quantitative climate
information from marine and terrestrial pollen archives, we focus on two key time intervals, the
early to mid-Holocene (8000 to 6000 cal yrs BP) and the late Holocene (4000 to 2000 yrs BP),
in order to test the above mentioned hypotheses on a Mediterranean-wide scale. Palynologically
derived climate information is compared with the output of regional-scale climate-model
simulations for the same time intervals.
Quantitative pollen-based precipitation estimates were generated along a longitudinal gradient
from the Alboran (West) to the Aegean Sea (East); they are derived from terrestrial pollen
records from Greece, Italy and Malta as well as from pollen records obtained from marine cores.
Because seasonality represents a key parameter in Mediterranean climates, special attention
was given to the reconstruction of season-specific climate information, notably summer and
winter precipitation. The reconstructed climatic trends corroborate a previously described
north-south partition of precipitation regimes during the Holocene. During the early Holocene,
relatively wet conditions occurred in the south-central and eastern Mediterranean region, while
drier conditions prevailed from 45°N northwards. These patterns reversed during the late
Holocene, with a wetter northern Mediterranean region and drier conditions in the east and
south. More sites from the northern part of the Mediterranean basin are needed to further
substantiate these observations. With regard to the existence of a west-east precipitation dipole
during the Holocene, our pollen-based climate data show that the strength of this dipole is
strongly linked to the seasonal parameter reconstructed: Early Holocene summers show a clear
east-to-west gradient, with summer precipitation having been highest in the central and eastern
Mediterranean and lowest over the western Mediterranean. In contrast, winter precipitation
signals are less spatially coherent. A general drying trend occurred from the early to the late
Holocene; particularly in the central and eastern Mediterranean. However, summer
precipitation in the east remained above modern values, even during the late Holocene interval.



Pollen-inferred precipitation estimates were compared to regional-scale climate modelling
simulations based on the HadAM3 GCM coupled to the dynamic HadSM3 and the high-
resolution regional HadRM3 models. Climate model outputs and pollen-inferred precipitation
estimates show remarkably good overall correspondence, although many simulated patterns are
of marginal statistical significance.  Nevertheless, models weakly support an east to west
division in summer precipitation and there are suggestions that the eastern Mediterranean
experienced wetter summer and winter conditions during the early Holocene and wetter
summer conditions during the late Holocene. The extent to which summer monsoonal
precipitation may have existed in the southern and eastern Mediterranean during the mid-
Holocene remains an outstanding question; our model, consistent with other global models,
does not suggest an extension of the African monsoon into the Mediterranean. Given the
difficulty in modelling future climate change in Southern Europe, more simulations based on
high resolution global models and very high resolution regional downscaling, perhaps even
including transient simulations, are required to fully understand the patterns of change in winter
and summer circulation patterns over the Mediterranean region.



## 1   Introduction


The Mediterranean region is particularly sensitive to climate change due to its position within
the confluence of arid North African (i.e., subtropically influenced) and temperate/humid
European (i.e., mid-latitudinal) climates (Lionello, 2012). Palaeoclimatic proxies, including
stable isotopes, lipid biomarkers, palynological data and lake-levels, have shown that the
Mediterranean region experienced climatic conditions that varied spatially and temporally
throughout the Holocene (e.g. Bar-Matthews and Ayalon, 2011; Luterbacher et al., 2012;
Lionello, 2012; Triantaphyllou et al., 2014, 2016; Mauri et al., 2015; De Santis and Caldara
2015; Sadori et al., 2016a) and well before (eg. Sadori et al., 2016b). Clear spatial climate
patterns have been identified from east to west and from north to south within the basin (e.g.
Zanchetta et al., 2007 ; Magny et al., 2009b, 2011, 2013; Zhornyak et al., 2011 ; Sadori et al.,
2013; Fletcher et al., 2013 ). Lake-level reconstructions from Italy suggest contrasting patterns
of palaeohydrological changes for the central Mediterranean during the Holocene (Magny et
al., 2012, 2013). Specifically, lake level maxima occurred south of approximately 40ºN in the
early to mid-Holocene, while lakes north of 40°N recorded minima. This pattern was reversed
at around 4500 cal yrs BP.
Quantitative pollen-based precipitation reconstructions from sites in northern Italy indicate
humid winters and dry summers during the early to mid-Holocene, whereas southern Italy was
characterised by humid winters and summers; the N-S pattern reverses in the late Holocene,
with drier conditions at southern sites and wet conditions at northern sites. These findings
support a North–South partition for the central Mediterranean with regards to precipitation, and
also confirm that precipitation seasonality is a key parameter in the evolution of Mediterranean
climates (Peyron et al., 2013). The pattern of shifting N-S precipitation regimes has also been
identified for the Aegean Sea (Peyron et al., 2013). Taken together, the evidence from pollen
data and from other proxies covering the Mediterranean region suggest a climate response that
can be linked to a combination of orbital, ice-sheet and solar forcings (Magny et al., 2013).
An east-west pattern of climatic change during the Holocene is also observed in the
Mediterranean region (e.g., Combourieu Nebout et al., 1998; Geraga et al., 2010; Colmenero-
Hildago et al., 2002; Kotthoff et al., 2008; Dormoy et al., 2009; Finne et al., 2011; Roberts et
al., 2011, 2012; Luterbacher et al., 2012; Guiot and Kaniewski, 2015). A gradient of
precipitation or an east-west division during the Holocene is suggested by marine pollen records
(Dormoy et al., 2009), lake-level reconstructions (Magny et al., 2013) and speleothem isotopes



(Roberts et al. 2011); the east-west pattern of change has also been corroborated through a
Bayesian inverse modelling approach (Guiot and Kaniewski, 2015)
This study aims to reconstruct and evaluate N-S and W-E climate gradients for the
Mediterranean basin, over two key periods in the Holocene, 8000-6000 cal yrs BP, and 4000-
2000 cal yrs BP. We estimate the magnitude of precipitation changes and reconstruct climatic
trends across the Mediterranean using both terrestrial and marine high-resolution pollen
records. Precipitation is estimated using the Modern Analogue Technique (Guiot 1990) for five
pollen records from Greece, Italy and Malta, and for eight marine pollen records along a
longitudinal gradient from the Alboran Sea to the Aegean Sea. Because precipitation
seasonality is a key parameter of change during the Holocene in the Mediterranean (Rohling et
al., 2002; Peyron et al., 2011, Mauri et al., 2015), the quantitative climate estimates focus on
reconstructing changes in summer and winter precipitation.
Paleoclimate proxy data are essential benchmarks for model intercomparison and validation
(e.g., Morrill et al., 2012; Heiri et al., 2014). This holds particularly true considering that
previous model-data intercomparisons have revealed substantial difficulties for GCMs in
simulating key aspects of Holocene climate (Hargreaves et al., 2013) for Europe (Mauri et al.,
2014), and notably for Southern Europe (Davis and Brewer, 2009; Mauri et al., 2015). We aim
to identify and quantify the spatio-temporal climate patterns in the Mediterranean Basin for two
key intervals of the Holocene (8000–6000 and 4000–2000 cal yrs BP) based on terrestrial and
marine high-resolution pollen records. Spatially, we focus on transects across the
Mediterranean basin from north to south and from west to east. Because precipitation
seasonality is a key parameter of Holocene climate change in the Mediterranean (Rohling et al.,
2002; Peyron et al., 2011, Mauri et al., 2015), our quantitative climate estimates focus on
summer and winter precipitation. Finally, we compare our pollen-inferred climate patterns with
regional-scale climate model simulations (Brayshaw et al., 2011a) in order to critically assess
the potential of the model set-up used to reproduce Holocene climate variability.

**2    Sites, pollen records, and models**
The Mediterranean region is at the confluence of continental and tropical air masses.
Specifically, the central and eastern Mediterranean is influenced by monsoonal systems, while
the north-western Mediterranean is under stronger influence from mid-latitude climate regimes
(Lionello et al., 2006). Mediterranean winter climates are mostly dominated by storm systems



originating over the Atlantic. In the western Mediterranean, precipitation is predominantly
affected by the North Atlantic Oscillation (NAO), while several systems interact to control
precipitation over the northern and eastern Mediterranean (Giorgi and Lionello, 2008).
Mediterranean summer climates are dominated by descending high pressure systems that lead
to dry/hot conditions, particularly over the southern Mediterranean where climate variability is
strongly influenced by African and Asian monsoons (Alpert et al., 2006) with strong
geopotential blocking anomalies over central Europe (Giorgi and Lionello, 2008; Trigo et al.,

148 2006).

The palynological component of our study combines results from five terrestrial and eight
marine pollen records to provide broad coverage of the Mediterranean basin (Figure 1, Table
1). The terrestrial sequences comprise pollen records from lakes along a latitudinal gradient
from northern Italy (Lakes Ledro and Accesa) to Sicily (Lake Pergusa), one pollen record from
Malta (Burmarrad) and one pollen record from Greece (Tenaghi Philippon). The marine pollen
sequences are situated along a longitudinal gradient across the Mediterranean Sea; from the
Alboran Sea (ODP Site 976 and core MD95-2043), Siculo-Tunisian strait (core MD04-2797),
Adriatic Sea (core MD90-917), and Aegean Sea (cores SL152, MNB-3, NS14, HCM2/22). For
each record we used the chronologies as reported in the original publications (see Table 1 for
references).
Climate reconstructions for summer and winter precipitation (Figs. 2, 3) inferred from the
terrestrial sequences and marine pollen records were performed using the Modern Analogue
Technique (MAT; Guiot, 1990). The MAT compares fossil pollen assemblages to modern
pollen assemblages with known climate parameters. The MAT is calibrated using an expanded
surface pollen dataset with more than 3600 surface pollen samples from various European
ecosystems (Peyron et al., 2013). In this dataset, 2200 samples are from the Mediterranean
region, and the results shows that the analogues selected here are limited to the Mediterranean
basin. Since the MAT use the distance structure of the data and essentially perform local fitting
of the climate parameter (as the mean of $n$-closest sites) they may be less susceptible to
increased noise in the data set, and less likely to report spurious values than others methods (for
more details on the method, see Peyron et al., 2011). *Pinus* is overrepresented in marine pollen
samples (Heusser and Balsam, 1977; Naughton et al,. 2007), and as such *Pinus* pollen was
removed from the assemblages for the calibration of marine records using MAT.
Climate model simulations focused on regional-scale climate modelling simulations based on
the HadAM3 GCM and the high-resolution regional HadRM3 models. Climate simulations are



described fully in Brayshaw et al. (2010, 2011a, b). The HadAM3 global atmospheric model
(resolution 2.5º latitude x 3.75º longitude, 19 vertical levels; Pope et al., 2000) is coupled to a
slab ocean (Hewitt et al., 2001) and used to perform a series of time slice experiments. Each
time-slice simulation corresponds to 20 model years after spin up (40 model years for pre-
industrial). The time slices correspond to "preindustrial", 2000 cal BP, 4000 cal BP, 6000 cal
BP and 8000 cal BP conditions, and are forced with appropriate insolation (associated with
changes in the Earth's orbit), and atmospheric $CO_2$ and $CH_4$ concentrations. The heat fluxes in
the ocean are held fixed (and there is no sea-level change) using values taken from a pre-
industrial control run, but sea-surface temperatures are allowed to evolve freely. The coarse
global output from the model for each time slice is downscaled over the Mediterranean region
using HadRM3 (i.e. a limited area version of the same atmospheric model; resolution 0.44º x
0.44º, with 19 vertical levels). Unlike the global model, HadRM3 is not coupled to an ocean
model; instead, sea-surface temperatures are derived directly from the HadSM3 output.
To aid interpretability (and to increase the signal-to-noise ratio), time slice experiments are
grouped into "late Holocene" (4000 BP and 2000 cal yrs BP) and "mid Holocene" (8000 BP
and 6000 cal yrs BP) experiments. Changes in climate are expressed as differences with respect
to the preindustrial control run and statistical significance is assessed with the Wilcoxon-Mann-
Whitney significance test (Wilks, 1995).

### 193   3     Results and Discussion


*A North-South precipitation pattern?*
Proxy evidence shows contrasting patterns of palaeohydrological changes in the central
Mediterranean. The early-to-mid-Holocene was characterized by lake-level and precipitation
maxima south of around 40°N. At the same time, northern Italy experienced precipitation and
lake-levels minima. This pattern reverses after 4500 cal yrs BP (Magny et al., 2012b; Peyron et
al., 2013). Other proxies suggest contrasting North-South hydrological patterns across the
Mediterranean (Magny et al., 2013). We focus on two key time periods, the early to mid-
Holocene (8000-6000 cal yrs BP), and the late Holocene (4000-2000 cal yrs BP) in order to test
this hypothesis across the Mediterranean, and to compare the results with regional climate
simulations for the same time periods.
Early to mid-Holocene (8000 to 6000 cal yrs BP)





Climatic trends reconstructed from both marine and terrestrial pollen records seem to
corroborate the hypothesis of a north-south division in precipitation regimes during the
Holocene (Fig 2a). Our results confirm that northern Italy was characterized by drier conditions
(relative to modern) while the south-central Mediterranean experienced more annual, winter
and summer precipitation during the early to mid-Holocene (Fig. 2a). Only Burmarrad (Malta)
shows drier conditions in the early to mid-Holocene (Fig 2a), although summer precipitation
reconstructions are marginally higher than modern at the site. Wetter summer conditions in the
Aegean Sea suggest a regional, wetter, climate signal over the central and eastern
Mediterranean. Winter precipitation in the Aegean Sea is less spatially coherent, with dry
conditions in the North Aegean Sea and wet or near-modern conditions in the Southern Aegean
Sea (Fig. 2a).
Precipitation reconstructions are particularly important for this region given that precipitation
rather than temperature represents the dominant controlling factor on Mediterranean
environmental system during the early to mid-Holocene (Renssen et al., 2012). Pollen and non-
pollen proxies, including marine and terrestrial biomarkers (terrestrial n-alkanes), indicate
humid mid-Holocene conditions in the Aegean Sea (Triantaphyllou et al., 2014, 2016). Results
within the Aegean support the pollen-based reconstructions, but non-pollen proxy data are still
lacking at the basin scale in the Mediterranean, limiting our ability to undertake independent
evaluation of precipitation reconstructions.
Very few large-scale climate reconstruction of precipitation exist for the whole Holocene
(Bartlein et al., 2011; Mauri et al., 2014; Guiot and Kaniewski, 2015, Tarroso et al., 2016) and,
even at local scales, pollen-inferred reconstructions of seasonal precipitation are very rare (Wu
et al., 2007; Peyron et al., 2011, 2013; Combourieu-Nebout et al., 2013, Nourelbait et al., 2016).
Several studies focused on the 6000 cal years BP period: Wu et al. (2007) reconstruct regional
seasonal and annual precipitation and suggest that precipitation did not differ significantly from
modern conditions across the Mediterranean; however, scaling issues render it difficult to
compare their results with the reconstructions presented here. Cheddadi et al. (1997) reconstruct
wetter-than-modern conditions at 6000 yrs cal BP in southern Europe; however, their study uses
only one record from Italy and measures the moisture availability index which is not directly
comparable to precipitation *sensu stricto* since it integrates temperature and precipitation. At
6000 yrs cal BP, Bartlein et al. (2011) reconstruct Mediterranean precipitation at values between
100 and 500 mm higher than modern. Mauri et al. (2015), in an updated version of Davis et al.
(2003), provide a quantitative climate reconstructions comparable to the seasonal precipitation



reconstructions presented here. Compared to Davis et al. (2003), which focused on Holocene
pollen-based temperature reconstructions for Europe, Mauri et al. (2015) have a broader set of
sites and present reconstructed seasonal and annual precipitation. Mauri et al. (2015) results differ
from the current study in using MAT with plant functional type scores and in producing gridded
climate maps (Fig. 2b). Mauri et al. (2015) show wetter summers in Southern Europe (Greece
and Italy) with a precipitation maximum between 8000 and 6000 cal yrs BP (Fig 2b), where
precipitation was ~20 mm/month higher than modern. As in our reconstruction, precipitation
changes in the winter were small and not significantly different from present-day conditions (Fig
2b). Our reconstructions are in good agreement with Mauri et al. (2015), with summer (and
annual) precipitation lower than modern over the northern Mediterranean region and wetter
summer conditions over much of the south-central Mediterranean, while winter conditions
appear to be similar to modern values. Mauri et al. (2015) results inferred from terrestrial pollen
records and the climatic trends reconstructed here from marine and terrestrial pollen records
seems to corroborate the hypothesis of a north-south division in precipitation regimes during the
Early to Mid-Holocene in central Mediterranean.

Late Holocene (4000 to 2000 cal yrs BP)
Late Holocene reconstructions of winter and summer precipitation indicate that the pattern
established during the early Holocene was reversed by 4000 cal yrs BP, with higher
precipitation in northern Italy and lower precipitation in southern Italy and Malta (Fig. 2a).
Annual precipitation reconstructions suggest drying relative to the early Holocene, with modern
conditions in northern Italy, and drier than modern conditions in central and southern Italy
during most of the Late Holocene. Reconstructions for the Aegean Sea indicate higher summer
and annual precipitation (Fig. 2b). Winter conditions reverse the early to mid-Holocene trend,
with wetter conditions in the northern Aegean Sea and drier conditions in the southern Aegean
Sea (Fig. 2b). Our reconstructions from all sites show a good fit with Mauri et al. (2015), except
for the Alboran Sea where we reconstruct relatively wet conditions, whereas Mauri et al. (2015)
reconstruct dry conditions (Fig. 2b). Our reconstruction of summer precipitation is very similar
to Mauri et al. (2015) for Greece and the Aegean Sea where wet conditions are reported (Fig.
2b).



*An East-West precipitation pattern?*
An East to West precipitation gradient, or an East-West division during the Holocene has been
suggested for the Mediterranean from pollen data and lakes isotopes (Dormoy et al., 2009;
Roberts et al., 2011; Guiot and Kaniewski, 2015). However, lake-levels and other hydrological
proxies around the Mediterranean Basin do not clearly support this hypothesis and rather show
contrasting hydrological patterns south and north of 40°N particularly during the Holocene
climatic optimum (Magny et al., 2013).
Early to mid-Holocene (8000 to 6000 cal yrs BP)
The annual precipitation and seasonal precipitation signals appear to conflict in the early
Holocene (Fig. 2a). The pollen-inferred annual precipitation indicates unambiguously wetter
than today conditions south of 45°N in the western, central and eastern Mediterranean, except
for Malta (Fig. 2a). Winter conditions show less spatial coherence, although the western basin
appears to have experienced higher precipitation than modern, while drier conditions exist in
the east (Fig. 2a). A prominent feature of the summer precipitation signal is an East to West
signature of increasing summer precipitation.
Our reconstruction shows a good match to Guiot and Kaniewski (2015) who have also discussed
a possible east-to-west division in the Mediterranean with regard to precipitation (summer and
annual) during the Holocene. They report wet centennial-scale spells in the eastern
Mediterranean during the Early Holocene (until 6000 years BP), with dry spells in the western
Mediterranean. Mid-Holocene reconstructions show continued wet conditions, with drying
through the late Holocene (Guiot and Kaniewski, 2015). This pattern indicates a see-saw effect
over the last 10,000 years, particularly during dry episodes in the Near and Middle East. As in
our findings, Mauri et al. (2015) also reconstruct high annual precipitation values over much of
the southern Mediterranean, and a weak winter precipitation signal. Mauri et al. (2015) confirm
an east-west gradient for summer precipitation, with conditions drier or close to present in
south-western Europe and wetter in the central and eastern Mediterranean (Fig 2b). These
studies corroborate the hypothesis of an east-to-west division in precipitation during the early
to mid-Holocene in the Mediterranean as proposed by Roberts et al. (2011). Roberts et al.
(2011) suggest the eastern Mediterranean (mainly Turkey and more eastern regions)
experienced higher winter precipitation during the early Holocene, followed by an oscillatory
decline after 6000 yrs BP. Our findings reveal wetter annual and summer conditions in the
eastern Mediterranean, although the winter precipitation signal is less clear. However, the



highest precipitation values reported by Roberts et al. (2011) were from sites located in western-
central Turkey; these sites are absent in the current study. Climate variability in the eastern
Mediterranean during the last 6000 years is documented in a number of studies based on
multiple proxies (Finné et al., 2011). Most palaeoclimate proxies indicate wet mid-Holocene
conditions (Bar-Matthews et al., 2003; Stevens et al., 2006; Eastwood et al., 2007; Kuhnt et al.,
2008; Verheyden et al., 2008) which agree well with our results; however most proxies are not
seasonally resolved.
Roberts et al. (2011) and Guiot and Kaniewski (2015) suggest that changes in precipitation in
the western Mediterranean were smaller in magnitude during the early Holocene, while the
largest increases occurred during the mid-Holocene, around 6000-3000 cal BP, before declining
to modern values. Speleothems from southern Iberia suggests a humid early Holocene (9000-
7300 cal BP) in southern Iberia, with equitable rainfall throughout the year (Walczak et al.,
2015). Our reconstructions for the Alboran Sea which clearly shows an amplified precipitation
seasonality (with higher annual/winter and lower than present summer rainfall) for the Alboran
sites. It is likely that seasonal patterns defining the Mediterranean climate must have been even
stronger in the early Holocene to support the wider development of sclerophyll forests than
present in south Spain (Fletcher et al., 2013).

Late Holocene (4000 to 2000 cal yrs BP)
Annual precipitation reconstructions suggest drier or near-modern conditions in central Italy
and Malta (Fig. 2b). In contrast, the Alboran and Aegean seas remain wetter. Winter and
summer precipitation produce opposing patterns: a clear east-west division exists for summer
precipitation, with a maximum in the eastern and a minimum over the western and central
Mediterranean (Fig. 2b). Winter precipitation shows the opposite trend, with a maximum in the
western Mediterranean and a minimum in the central and eastern Mediterranean (Fig. 2b). Our
results are also in agreement with lakes and speleothem isotope records over the Mediterranean
for the late Holocene (Roberts et al., 2011), and the Finné et al. (2011) palaeoclimate synthesis
for the eastern Mediterranean. There is a good overall correspondence between trends and
patterns in our reconstruction and that of Mauri et al. (2015), except for the Alboran Sea (Fig.
2b). High-resolution speleothem data from southern Iberia show Mediterranean climate
conditions in southern Iberia between 4800 and 3000 cal BP (Walczak et al., 2015) which is in
agreement with our reconstruction. The Mediterranean climate conditions reconstructed here



for the Alboran Sea during the late Holocene is consistent with a climate reconstruction
available from the Middle Atlas (Morocco), which show a trend over the last 6000 years
towards arid conditions as well as higher precipitation seasonality between 4000 and 2000 cal
yrs BP (Nourelbait et al., 2016). There is also good evidence from many records to support late
Holocene aridification in southern Iberia. Paleoclimatic studies document a progressive
aridification trend since ~7000 cal yr BP (e.g. Carrion et al., 2010; Jimenez-Moreno et al., 2015,
Ramos-Roman et al., 2016), although a reconstruction of the annual precipitation inferred from
pollen data with the Probability Density Function method indicate stable and dry conditions in
the south of the Iberian Peninsula between 9000 and 3000 cal BP (Tarroso et al., 2016).
The current study shows that a prominent feature of late Holocene climate is the east-west
division in precipitation, which varies based on the seasonal parameter reconstructed: summers
were overall dry or near-modern in the central and western Mediterranean and wetter in the
eastern Mediterranean, while winters were wet in the western Mediterranean and drier in the
central and eastern Mediterranean.

*Data-model comparison*
Figure 3 shows the data-model comparisons for the early to mid-Holocene (a) and late Holocene
(b) compared to present values (in anomalies). Encouragingly, there is a good overall
correspondence between patterns and trends in pollen-inferred precipitation and model outputs.
Caution is required when interpreting climate model results as many of the changes depicted in
Fig. 3 are very small and of marginal statistical significance, suggesting a high degree of
uncertainty around their robustness.
For the early to mid-Holocene, both model and data indicate wet annual, winter and summer
conditions in the Eastern Mediterranean. There are indications of an east to west division in
summer precipitation simulated by the climate model (the magnitude of the increase in the
eastern side of the basin is, however, extremely small). Furthermore, in the Aegean Sea, the
model shows a good match with pollen-based reconstructions, suggesting that the increased
spatial resolution of the regional climate model helps to simulate the localized, "patchy",
impacts of Holocene climate change, when compared to coarser global GCMs (Fig. 3). In Italy,
the model shows a good match with pollen-based reconstructions with regards to the contrasting
north-south precipitation regimes, but there is little agreement between model output and
climate reconstruction with regard to winter and annual precipitation in southern Italy. The



climate model suggests wetter winter and annual conditions in the far western Mediterranean
(i.e., western Iberia and the NW coast of Africa) – similar to pollen-based reconstructions – and
near-modern summer conditions during summers.
Model and pollen-based reconstructions for the late Holocene indicate declining winter
precipitation in the eastern Mediterranean and southern Italy (Sicily and Malta), although
model-based changes are not statistically significant. In contrast, late Holocene summer
precipitation is higher than today in the eastern Mediterranean (though only marginally so in
the climate model). The east-west division in summer precipitation is strongest during the late
Holocene and there are suggestions that it appears to be consistently simulated in the climate
model but again, the signal – particularly in the Eastern Mediterranean – is not statistically
significant.
Our findings are consistent with previous data-model comparisons based on the same regional
model. Previous comparisons suggested that the winter precipitation signal was strongest in the
northeastern Mediterranean (near Turkey) during the early Holocene (Brayshaw et al., 2011a;
Roberts et al., 2011) and that there was a drying trend in the Mediterranean from the early
Holocene to the late Holocene, particularly in the east. This is coupled with a gradually
weakening seasonal cycle of surface air temperatures towards the present.
In contrast to Holocene winter precipitation changes in the Mediterranean (which are consistent
with simulated changes in Mediterranean storm tracks; Brayshaw et al 2010), it is clear that
most global climate models (PMIP2, PMIP3) simulate only very small changes in summer
precipitation in the Mediterranean during the Holocene (Braconnot et al., 2007a,b, 2012; Mauri
et al., 2014). The lack of a summer precipitation signal is consistent with the failure of the north-
eastern extension of the west African monsoon to reach the southeastern Mediterranean, even
in the early-to-mid-Holocene (Brayshaw et al., 2011a). Even though the regional climate model
simulates a small change in precipitation compared to the proxy results, it cannot be robustly
identified as statistically significant. This is to some extent unsurprising, insofar as the regional
climate simulations presented here are themselves "driven" by data derived from a coarse global
model (which, like its PMIP2/3 peers, does not simulate an extension of the African monsoon
into the Mediterranean during this time period). Therefore, questions about summer
precipitation in the Eastern Mediterranean during the Holocene remain. Climate dynamics need
to be better understood in order to confidently reconcile proxy data (which suggest increased
summer precipitation during the early Holocene in the Eastern Mediterranean) with climate
model results. Based on the high-resolution coupled climate model EC-Earth, Bosmans et al.



(2015) shows how the seasonality of Mediterranean precipitation should vary from minimum
to maximum precession, indicating a reduction in precipitation seasonality, due to changes in
storm tracks and local cyclogenesis (*i.e.*, no direct monsoon required). Such high-resolution
climate modeling studies (both global and regional) may prove a key ingredient in simulating
the relevant atmospheric processes (both local and remote) and providing fine-grain spatial
detail necessary to compare results to palaeo-proxy observations.
Future work based on transient Holocene model simulations are important, nevertheless,
transient-model simulations have also shown mid-Holocene data-model discrepancies (Fischer
and Jungclaus, 2011; Renssen et al., 2012). It is, however, suggested that further work is
required to fully understand changes in winter and summer circulation patterns over the
Mediterranean (Bosmans et al., 2015).

*Limitations*
Classic ecological works for the Mediterranean (e.g. Ozenda 1975) highlight how precipitation
limits vegetation type in plains and lowland areas, but temperature gradients take primary
importance in mountain systems. Also, temperature and precipitation changes are not
independent, but interact through bioclimatic moisture availability and growing season length
(Prentice et al., 1996). This may be one reason why certain sites diverge from model outputs:
the Alboran sites, for example, integrate pollen from the coastal plains through to mountain
(+1500m) elevations. At high elevations within the source area, temperature effects become be
more important than precipitation in determining the forest cover type. So, it will not be possible
to fully isolate precipitation signals from temperature changes. Particularly for the semiarid
areas of the Mediterranean, the reconstruction approach probably cannot distinguish between a
reduction in precipitation and an increase in temperature and PET, or vice versa.
Along similar lines, while the concept of reconstructing winter and summer precipitation
separately is very attractive, it may be worth openly commenting on some limitations. Although
different levels of the severity or length of summer drought are an important ecological
limitation for vegetation, reconstructing absolute summer precipitation can be difficult as the
severity/length of bioclimatic drought is determined by both temperature and precipitation.
Also, we are dealing with a season which has, by definition, small amounts of precipitation that
drop below the requirements for vegetation growth. Elevation is also of concern, as lowland
systems tend to be recharged by winter rainfall, but high mountain systems may receive a



significant part of precipitation as snowfall, which is not directly available to plant life. This may be important in the long run for improving the interpretation of long-term Holocene changes and contrasts between different proxies, such as lake-levels and speleothems. All of these points may seem very picky on the ecology side, but they may have a real influence leading to problems and mismatches between different reconstruction approaches and different proxies (e.g. Davis et al., 2003; Mauri et al., 2015).

Another important point is the question of human impact on the Mediterranean vegetation during the Holocene. Since human activity has influenced natural vegetation, distinguishing between vegetation change induced by humans and climatic change in the Mediterranean is a challenge requiring independent proxies and approaches. Therefore links and processes behind societal change, and climate change in the Mediterranean region increasingly being investigated (eg. Holmgren et al., 2016; Gogou et al, 2016; Sadori et al., 2016a). Here, the behavior of the reconstructed climatic variables between 4000 and 2000 cal yrs BP is likely to be influenced by non-natural ecosystem changes due to human activities such as the forest degradation that began in lowlands, progressing to mountainous areas (Carrión et al., 2010). These human impacts add confounding effects for fossil pollen records and may lead to slightly biased temperature reconstructions during the Late Holocene, likely biased towards warmer temperatures and lower precipitation. However, if human activities become more marked at 3000 cal ky BP, they increase significantly over the last millennia (Sadori et al., 2016) which is not within the time scale studied here. Moreover there is strong agreement between summer precipitation and independently reconstructed lake-level curves (Magny et al., 2013). For the marine pollen cores, human influence is much more difficult to interpret given that the source area is so large, and that, in general, anthropic taxa are not found in marine pollen assemblages.

**Conclusions**

The Mediterranean is particularly sensitive to climate change but the extent of future change relative to changes during the Holocene remains uncertain. Here, we present a reconstruction of Holocene precipitation in the Mediterranean using an approach based on both terrestrial and marine pollen records, along with a model-data comparison. We investigate climatic trends across the Mediterranean during the Holocene to test the hypothesis of an alternating north-south precipitation regime, and/or an east-west precipitation dipole. We give particular emphasis to the reconstruction of seasonal precipitation considering the important role it plays in this system.



Climatic trends reconstructed in this study seem to corroborate the north-south division of
precipitation regimes during the Holocene, with wet conditions in the south-central and eastern
Mediterranean, and dry conditions above 45°N during the early Holocene, while the opposite
pattern dominates during the late Holocene. This study also shows that a prominent feature of
Holocene climate in the Mediterranean is the east-to-west division in precipitation, strongly
linked to the seasonal parameter reconstructed. During the early Holocene, we observe an east-
to-west division with high summer precipitation in the central and eastern Mediterranean and a
minimum over the western Mediterranean, while the signal for winter precipitation is less
spatially consistent. There was a drying trend in the Mediterranean from the early Holocene to
the late Holocene, particularly in central and eastern regions but summers in the east remained
wetter than today.
The regional climate model outputs show a remarkable qualitative agreement with our pollen-
based reconstructions, though it must be emphasised that the changes simulated are typically
very small and of questionable statistical significance. Nevertheless, there are indications that
the east to west division in summer precipitation reconstructed from the pollen records do
appear to be simulated by the climate model. The model results also suggest that parts of the
eastern Mediterranean experienced wetter conditions both in winter and in summer during the
early and late Holocene and marginally wetter conditions in summer during the late Holocene
(both consistent with the paleo-records). It is therefore noted that the use of higher-resolution
climate models (both regional and global) may offer benefits for data-model comparison: both
due to the inherently "patchy" nature of climate signals and palaeo-records, and through the
better representation of the underlying atmospheric dynamics. It is therefore argued that more
model simulations – ideally with higher resolution atmospheric dynamics – are required to fully
understand the changes in the winter and summer circulation patterns over the Mediterranean
region.

**Acknowledgements**
This study is a part of the LAMA ANR Project (MSHE Ledoux, USR 3124, CNRS)
financially supported by the French CNRS (National Centre for Scientific Research). Simon
Goring is currently supported by NSF Macrosystems grant 144-PRJ45LP. This is an ISEM
contribution n°XXXX.



**Figure captions**


Figure 1: Locations of terrestrial and marine pollen records along a longitudinal gradient from
west to east and along a latitudinal gradient from northern Italy to Malta. Ombrothermic
diagrams are shown for each site, calculated with the NewLoclim software program and
database, which provides estimates of average climatic conditions at locations for which no
observations are available (ex.: marine pollen cores).
Figure 2:
(a) Pollen-inferred climate estimates as performed with the Modern Analogues Technique
(MAT): annual precipitation, winter precipitation (winter = sum of December, January
and February precipitation) and summer precipitation (summer = sum of June, July
and August precipitation). Changes in climate are expressed as differences with
respect to the modern values (anomalies, mm/day). The modern values are derived
from the ombrothermic diagrams (cf fig. 1). Two key intervals of the Holocene
corresponding to the two time slice experiments (fig. 3) have been chosen: 8000–6000
and 4000–2000 cal yrs BP. The climate values available during these periods have
been averaged (stars).
(b) Comparison of our pollen-based climate reconstructions for the Mediterranean region with
the pollen-inferred climate reconstruction at the European scale of Mauri et al (2015),
expressed in anomaly (mm/month). These authors used the MAT with a modern
analogue selection based on PFT (plant functional type) scores (and not pollen
assemblages like the method used in this paper) and a 4D interpolation technique to
produce gridded paleoclimate maps (for more details, see Mauri et al., 2015).
Figure 3: Data-model comparison for mid and late Holocene precipitation, expressed in
anomaly (mm/day). Simulations are based on a regional model (Brayshaw et al., 2010):
standard model HadAM3 coupled to HadSM3 (dynamical model) and HadRM3 (high-
resolution regional model). The plots are hatched where it passes a significance test (threshold
used here 70%). Pollen-inferred climate estimates (stars) are the same as in Figure 2: annual
precipitation, winter precipitation (winter = sum of December, January and February
precipitation) and summer precipitation (summer = sum of June, July and August
precipitation).

Table 1: Metadata for the terrestrial and marine pollen records evaluated.



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

Contributors Data: Circum-Mediterranean fire activity and climate changes during the mid
Holocene environmental transition (8500-2500 cal yr BP), The Holocene, 21, 53-73, 2011.
Vannière, B., Magny, M. , Joannin, S. , Simonneau, A. , Wirth, S.B. , Hamann, Y., Chapron,
E., Gilli, A., Desmet, M., and Anselmetti, F.S.: Orbital changes, variation in solar activity and
increased anthropogenic activities: controls on the Holocene flood frequency in the Lake
Ledro area, Northern Italy, Clim. Past, 9, 1193-1209, 2013.
Verheyden S., Nader F.H., Cheng H.J., Edwards L.R. and Swennen R.: Paleoclimate
reconstruction in the Levant region from the geochemistry of a Holocene stalagmite from the
Jeita cave, Lebanon. Quaternary Research, 70, 368-381, 2008.
Walczak, I.W., Baldini, J.U.L., Baldini, L.M., Mcdermott, F., Marsden, S., Standish, C.D,
Richards, D.A., Andreo, B and Slater J.: Reconstructing high-resolution climate using CT
scanning of unsectioned stalagmites: A case study identifying the mid-Holocene onset of the
Mediterranean climate in southern Iberia, Quaternary Science Reviews 127, 117-128, 2015.
Wilks D. S.: Statistical methods in the atmospheric sciences (Academic Press, San Diego,
CA), 1995.
Wood, S.N. Fast stable restricted maximum likelihood and marginal likelihood estimation of
semiparametric generalized linear models. Journal of the Royal Statistical Society (B) 73(1),

851    3-36, 2011.

Wu, H., Guiot, J., Brewer, S., and Guo, Z.: Climatic changes in Eurasia and Africa
at the Last Glacial Maximum  and  mid-Holocene: reconstruction from pollen data using
inverse vegetation modelling, Clim. Dyn., 29, 211-229, 2007.
Zanchetta, G., Borghini, A., Fallick, A.E., Bonadonna, F.P., and Leone, G.: Late Quaternary
palaeohydrology of Lake Pergusa (Sicily, southern Italy) as inferred by stable isotopes of
lacustrine carbonates, J. Paleolimnol., 38, 227-239, 2007.
Zhornyak, L.V., Zanchetta, G., Drysdale, R.N., Hellstrom, J.C., Isola, I., Regattieri, E.,
Piccini, L., Baneschi, I., and Couchoud, I.: Stratigraphic evidence for a "pluvial phase"
between ca. 8200-7100 ka from Renella cave (Central Italy), Quat. Sci. Rev., 30, 409-417,

861    2011.



Figure 1: Locations of terrestrial (red) and marine (yellow) pollen records.
Ombrothermic diagrams are calculated with the NewLoclim software,
which provides estimates of average climatic conditions at locations for
which no observations are available (ex.: marine pollen cores).





Figure 2: 8000-6000 cal years BP
(A) Pollen-inferred climate estimates as performed with the Modern Analogues Technique : annual precipitation, winter precipitation (winter = sum of December, January and February precipitation) and summer precipitation (summer = sum of June, July and August precipitation). Changes in climate are expressed as differences with respect to the modern values (anomalies, mm/day) which are derived from the ombrothermic diagrams (cf fig. 1). Climate values reconstructed during the 8000-6000 cal yrs BP have been averaged (stars).
(B) Pollen-inferred climate reconstruction at the European scale of Mauri et al (2015), expressed in anomaly (mm/month). These authors used a modern analogue selection based on PFT (plant functional type) scores (and not pollen assemblages like the method used in A) and a 4D interpolation technique to produce gridded paleoclimate maps (for more details, see Mauri et al., 2015).



Figure 2: 4000-2000 cal yrs BP




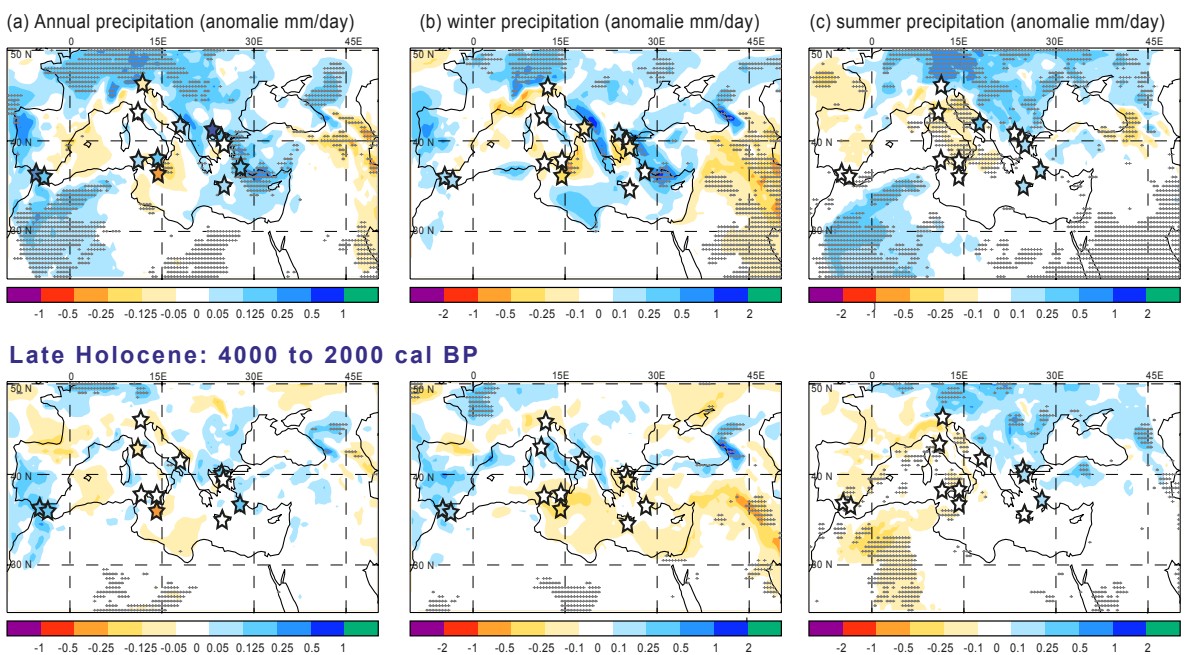

Figure 3: Data-model comparison for mid and late Holocene precipitation, expressed in anomaly (mm/day). Simulations are based on a regional model (Brayshaw et al., 2010): standard model HadAM3 coupled to HadSM3 (dynamical model) and HadRM3 (high-resolution regional model). The plots arehatched where it passes a significance test (threshold used here 70%).

Pollen-inferred climate estimates (stars) are the same as in Figure 2: annual precipitation, winter precipitation and summer precipitation .





**Terrestrial pollen records**

| | Longitude | Latitude | Elevation (m a.s.l) | References |
|---|---|---|---|---|
| **Lago di Ledro** (Northern Italy) | 10°76'E | 45°87'N | 652 | Joannin et al. (2013), Magny et al. (2009, 2012a), Vannière et al. (2013), Peyron et al. (2013) |
| **Accesa** (Central Italy) | 10°53'E | 42°59'N | 157 | Drescher-Schneider et al. (2007), Magny et al. (2007, 2013), Colombaroli et al. (2008), Sadori et al. (2011), Vannière et al. (2011), Peyron et al. (2011, 2013) |
| **Trifoglietti** (southern Italy) | 16°01'E | 39°33'N | 1048 | Joannin et al., (2012) ; Peyron et al. (2013) |
| **Pergusa** (Sicily) | 14°18'E | 37°31'N | 667 | Sadori and Narcisi (2001); Sadori and Giardini (2007); Sadori et al. (2008, 2011, 2013, 2016b); Magny et al. (2011, 2013) |
| **Tenaghi Philippon** (Greece) | 24°13.4'E | 40°58.4'N | 40 | Pross et al., (2009, 2015); Peyron et al. (2011); Schemmel et al., (2016) |
| **Burmarrad** (Malta) | 14°25'E | 35°56'N | 0.5 | Djamali et al., (2013); Gambin et al., (2016) |

**Marine pollen records**

| | Longitude | Latitude | Water-depth | References |
|---|---|---|---|---|
| **ODP 976** (Alboran Sea) | 4°18'W | 36°12' N | 1108 | Combourieu-Nebout et al., (1999, 2002, 2009) ; Dormoy et al., (2009) |
| **MD95-2043** (Alboran Sea) | 2°37'W | 36°9'N | 1841 | Fletcher and Sánchez Goñi( 2008); Fletcher et al., (2010) |
| **MD90-917** (Adriatic Sea) | 17°37'E | 41°97'N | 845 | Combourieu-Nebout et al., (2013) |
| **MD04-2797** (Siculo-Tunisian strait) | 11°40'E | 36°57'N | 771 | Desprat et al., (2013) |
| **SL152** (North Aegean Sea) | 24°36' E | 40°19' N | 978 | Kotthoff et al., (2008, 2011), Dormoy et al., (2009). |
| **NS14** (South Aegean Sea) | 27°02.87'E | 36°38.9'N | 505 | Kouli et al., (2012) ; Gogou et al., (2007); Triantaphyllou et al., (2009a, b) |
| **HCM2/22** (south Crete) | 24°53'E | 34°34 N | 2211 | Kouli et al., (2012) ; Triantaphyllou et al,( 2014) |
| **MNB-3** (North Aegean Sea) | 25°00'E | 39°15.43'N | 800 | Kouli et al., (2012) ; Triantaphyllou et al, (2014) |

Table 1: Metadata for the terrestrial and marine pollen records evaluated.