# Peer review of "Precipitation changes in the Mediterranean basin during"

_Climate of the Past, 2016_

## Referee Comment (RC1) · Anonymous Referee #1 · 21 Jul 2016

The manuscript presents a new compilation of proxy data based on pollen assemblages in the Northern Mediterranean basin covering the Holocene period. The analysis is focused on reconstructed seasonal and total precipitation changes to ascertain the existence of a North-South and west-east dipoles The results derived from the analysis of proxy data are compared with time-slice simulations with a regional atmosphere model driven at the boundaries by a global atmosphere model coupled to a slab ocean for different periods through the Holocene. My recommendation is to include revisions in the paper, which could be considered as between major and minor.

Although I do think that the topic of research is very interesting and this paper is a contribution to some of the research goals, I had the nagging feeling that the added

value of the study is difficult to grasp. Quite often, the authors conclude that their analysis confirms previous studies, mainly by Mauri et al., Guiot et al, Roberts et al. and others , but it is not easy to identify what are the new conclusions, or what new information the data or the analysis is providing beyond of what is already new.

This applies also to the model set-up. Mauri et al already compared pollen-based reconstruction with model output for the mid-Holocene. They used a full model ensemble, albeit global models with coarser spatial resolution, in contrast to the present study that uses a regional model but with a slap ocean. In the model set-up used by the authors there are some open questions. For instance, they use a slab ocean that ignores the ocean dynamics, but are the simulated sea-surface temperatures comparable to the temperatures simulated in global couple simulations for the mid-Holocene? what could be the role of the dynamics of the North Atlantic in determining the precipitation patterns in Europe? I am aware that a full coupled simulation over the Holocene could be out of the scope of the present study in terms of computer resources, but some type of validation or discussion of the possible shortcoming of the simulation set-up should be addressed. More importantly, I think, would be to identify which aspects of the regional modelling provide an added value relative to the global model results presented in of Mauri et al. . The manuscript includes just a comment in passing about the heterogeneity of simulated precipitation change sin the Balkans, but this is not really followed trough. For instance, one of the mechanisms that may explain the pattern or precipitation changes are shifts in the North Atlantic storm tracks. Is the regional model able to represent the storm tracks more realistically than the global models ? Is the representation of present-day precipitation better in the regional model than in the ensemble of CMIP5 global models ? I would assume the answer is yes, but it it would be nice to see it discussed in the manuscript as well. On the other hand, the slab ocean is likely not able to realistically represent the meridional sea-surface temperatures in the Atlantic. This may affect the intensity and extent of the African Monsoon and its changes over the Holocene. Could this limitation influence the simulation of t summer precipitation changes in the Mediterranean? All in all, the manuscript looks for these and similar

reasons more descriptive than it should and could be.

The conclusions could be presented in a more clear way. After reading this section several times, it remains unclear to me weather they precipitation dipoles really exists. Some sentence clearly say yes, but they are immediately qualified with a 'however,' or 'but. This is particular apparent in the paragraph starting in line 464 . The last paragraph in the conclusions looks also quite convoluted, and some of the conclusions are not really based on the results presented here. For instance, the authors conclude that the regional model represents better the atmospheric dynamics, and therefore precipitation. This can be somehow expected, but it has not been shown in this study, and in particular, it has not been shown that the particular model set-up used here ( with a slab ocean ) is indeed better.

The title is a bit misleading, as the study is basically about precipitation changes and not 'climate' in general.

Abstract has many elements of introduction, including recommendations, like the usa of transient simulations, which turn out to be correct but that they are not really substantiated by the results described in this manuscript. My criticisms is to some extent a matter of taste, but I think that the abstract should be succinct and mainly describing the methods, results and conclusions. Introductory remarks should go in the introduction, and final speculations or recommendations, in the main text

Lines 125-132 This is a repetition of a previous paragraph on the same page

It may be interesting to know the time resolution of the proxy records

line 239 I think that the reference to Mauri et al (2015) is not correct. First, year should be 2014 and not 2015. But, secondly, I think the authors are referring to another paper by Mauri (2015): The climate of Europe during the Holocene: a gridded pollen-based reconstruction and its multi-proxy evaluation. Quaternary Science Reviews 112 (2015) 109e127. The paper in climate of the past is an analysis of mid-Holocene conditions

whereas the QSR paper is an analysis of the evolution throughout the Holocene.

line 242 Mauri et al used a reconstruction method based on plant functional types. Should the reader expect differences to the reconstruction method used here ? Could some of the differences to the present results due to the different methodology ?

line 266 'Mediterranean, and dry conditions above 45°N during the early Holocene, while the opposite' North of 45N

Caption Figure 3. which is the reference period to calculate the simulated precipitation anomalies ?
* * *

---

## Referee Comment (RC2) · Anonymous Referee #2 · 6 Aug 2016

Review of "The climate of the Mediterranean basin during the Holocene from terrestrial and marine pollen records: A model/data comparison" by Peyron et al.

The paper presents climate reconstructions based on a set of pollen sequences in the Mediterranean basin for two time periods (early/late Holocene). They use the results to explore spatial patterning of past climate change, and further compare these to regional climate model output. The study is well set up, with very good background and the results are clearly displayed. I struggled, however, in understanding what the message of the paper was. The discussion is focused on the two possible spatial patterns (North-South and East-West), and on a comparison with the model output, but I didn't feel that these were really pulled together. If the goals was to understand

these patterns, then I feel that there is a missed opportunity in using the model output to understand the atmospheric drivers of the changes in spatial pattern. If the goal is purely a comparison, then the discussion on spatial patterning is much less relevant, other than as a benchmark. I would like to see the goals more clearly stated, and clearly referred to throughout the paper.

I would like to see more discussion about the choices of spatial pattern and of time period. I understand that the goal is to show both a latitudinal and longitudinal gradient, but the pattern often looks more complex. I was hoping that one outcome here would be a more synoptic view of the Mediterranean Holocene climate, considering the entire spatial pattern and the potential drivers of this. For the time period, I don't really understand why the authors did not look across the entire Holocene, but instead focused on two, quite long time periods. Are these gradients only a feature of the time periods chosen? What was the variation outside (or even within) these periods? Given that one of the papers they cite has already completed full Holocene reconstructions (Mauri et al., 2015), and that there is interest in full Holocene/Glacial transient GCM simulations, this snapshot approach appears to be somewhat limited. At the very least, it would be good to have a better justification for the choices than "to aid interpretability".

The choice of precipitation as a variable for comparison also needs better justification. The authors state (line 416) that using precipitation instead of moisture indices may be why there is a model/data mismatch, and some form of moisture index has been proposed as a better quantity for pollen reconstructions elsewhere (Bartlein et al., 2011). Given this, and that alpha is routinely reconstructed from pollen, why not use this instead? And if not, please justify the use of precipitation, given its limitation as a reconstructed variable.

Minor comments follow:

Line 28. The abstract could be shortened and made more concise – there is some repetition (e.g. lines 39-40 and lines 60-61)

[Figure]

Line 34 (and elsewhere). Is the pattern a gradient or dipole? These are not to my understanding the same thing, as one represents a trend, and the other represents a pattern of two opposing centers. Please either use one or the other, or state more clearly which is being referred to at any time.

Line 38-40. What is the aim of the comparison?

Lines 47-51. This section needs some rewriting to make it clear when the authors are referring to conditions being drier in one region than another, or that the anomalies are drier compared to another time period.

Line 61. In what sense is HadSM3 dynamic? (and what is HadSM3 as opposed to the other models shown here)

Lines 90-92. Needs a citation

Line 93. Which sites in N. Italy? Citation, please.

Lines 126-127. Why these periods? Why are they 'key'? Why not do this in a continuous way?

Lines 133-134. "To critically assess the potential of the model setup…" This is a little fuzzy, but I assume that the goal is to discuss the regional climate model output, and the model parameters. However, I don't really feel that this was addressed in the discussion. There is some discussion of findings in other papers (e.g. Bosmans et al) but nothing about the setup used here.

Line 169. Arguably, pine is overrepresented in all sites. Why only exclude it for the marine sites? How big an impact does this have?

Line 187. I think I get what the authors are saying here, but given the increasing interest in transient simulations (e.g. Liu et al) and reconstructions (e.g. Marcotte et al), I'd like to see this choice justified a little better

Line 190. The model uses the pre-industrial period as a baseline for anomalies,

whereas I assume the pollen reconstructions use the late 20th century, although this is not specified. How much will this affect the offset between model and data. How big are the reconstructed anomalies relative to any change between these periods?

Line 206. The results are more single points in time, rather than climate trends

Lines 217-219. This seems like it would be more appropriate in the introduction

Line 231. What scaling issues?

Line 247. It is difficult to see visually how there is good corroboration between thes results and Mauri et al. Would it be possible to carry out a one-to-one comparison of values, and test the differences? Further – as both Mauri et al, and this study present statistical climate reconstructions from pollen data, it is hard to see how the agreement between them supports the robustness of the results.

Line 351. How can you tell that the data-model agreement is good? Again, some point-by-point comparison would help (and help highlight where the main differences are)

Line 435. How would the snowpack affect different methods? I can see that it might affect proxies differently, but not methods.

Line 486. It is hard to disagree with a call for higher resolution in climate models, but how exactly will this help? What processes will be better represented, allowing for better climate simulation?

---

## Editor Comment (EC1) · J. Guiot (Editor) · 28 Aug 2016

Dear authors,

We have received two reviews of your paper. You can see that the main criticism concerns the difficult to understand the goals of the paper and its added value. They think that there is potential in this paper but you have some work to do to clarify these points. You have now to post replies to all the comments, and to prepare a revised version of your paper accordingly. In the revised version, I would like to see your corrections in track change mode.

Best regards

[Figure]

Joel Guiot

---

## Author Comment (AC1) · 18 Nov 2016

Paper: The climate of the Mediterranean basin during the Holocene from terrestrial and marine pollen records: A model/data comparison

By Odile Peyron et al

Clim. Past Discuss: cp-2016-65

Reply to the reviewers' comments An important point came out in Review 2 (point 15), concerning the use of the preindustrial baseline. In the model simulations, we have always used PREIND as the baseline because the climate forcing before then, over the Holocene, is mostly orbital; in contrast to the industrial period where it is mostlygreenhouse gas. It does, however, have some impact on the precipitation signals we are discussing here (new figure 4). Therefore, in this revised version, we have changed our model-data synthesis (Fig. 3) and have taken present day in the control run instead of preindustrial to be in better agreement with the pollen data (the pollen data precipitation is best seen as 'anomalies relative to present day 1960-1990').

First reviewer 1) Quite often, the authors conclude that their analysis confirms previous studies, but it is not easy to identify what are the new conclusions, or what new information the data or the analysis is providing beyond of what is already new. Text in the Introduction part has been changed to: "The first originality of our approach is that we estimate the magnitude of precipitation changes and reconstruct climatic trends across the Mediterranean using both terrestrial and marine high-resolution pollen records. The signal reconstructed is then more regional than in the studies based on terrestrial records alone. Moreover, this study aims to reconstruct precipitations patterns for the Mediterranean basin over two key periods in the Holocene, while the existing large-scale quantitative paleoclimate reconstructions for the Holocene are often limited to the mid-Holocene - 6000 yrs BP- (Cheddadi et al., 1997; Bartlein et al., 2011; Mauri et al., 2014), except the climate reconstruction for Europe proposed by the study of Mauri et al. (2015). The second originality of our approach is that we propose a data/model comparison based on: (1) two time-slices and not only the mid-Holocene, a standard benchmark time period for this kind of data–model comparison; (2) a high resolution regional model (RCM) which provides a better representation of local/regional processes and helps to better simulate the localized, "patchy", impacts of Holocene climate change, when compared to coarser global GCMs (e.g. Mauri et al., 2014); (3) changes in seasonality, particularly changes in summer atmospheric circulation which have not been widely investigated (Brayshaw et al., 2011)."

2) In the model set-up used by the authors there are some open questions. For instance, they use a slab ocean that ignores the ocean dynamics, but are the simulated sea-surface temperatures comparable to the temperatures simulated in global couple

simulations for the mid-Holocene? what could be the role of the dynamics of the North Atlantic in determining the precipitation patterns in Europe? I am aware that a full coupled simulation over the Holocene could be out of the scope of the present study in terms of computer resources, but some type of validation or discussion of the possible shortcoming of the simulation set-up should be addressed. More importantly, I think, would be to identify which aspects of the regional modelling provide an added value relative to the global model results presented in of Mauri et al. . The manuscript includes just a comment in passing about the heterogeneity of simulated precipitation changes in the Balkans, but this is not really followed trough. For instance, one of the mechanisms that may explain the pattern or precipitation changes are shifts in the North Atlantic storm tracks. Is the regional model able to represent the storm tracks more realistically than the global models ? Is the representation of present-day precipitation better in the regional model than in the ensemble of CMIP5 global models ? I would assume the answer is yes, but it it would be nice to see it discussed in the manuscript as well. On the other hand, the slab ocean is likely not able to realistically represent the meridional sea-surface temperatures in the Atlantic.This may affect the intensity and extent of the African Monsoon and its changes over the Holocene. Could this limitation influence the simulation of t summer precipitation changes in the Mediterranean?

This comment raises several important issues, which we attempt to disentangle as follows.

We agree with the reviewer that there are indeed limitations in the climate modeling approach used here. These were discussed at length in previous publications, as cited from the present article. Brayshaw et al., 2010 (Phil Trans A), in particular, goes into detail on (A) evaluating the relative merits and difficulties of the modeling approach compared to others (such as PMIP), including the role of embedded high-resolution regional modelling and (B) discussing the physical atmospheric drivers of winter-time change such as storm tracks, Hadley cell expansion/contraction and teleconnections from the Indian Ocean. Brayshaw et al., 2011 (Holocene) provides a wider review (both summer and winter), discussing of the impact of the GCM's simulation of tropical Atlantic SSTs (on, e.g., the summer expansion of the African monsoon) as suggested by the reviewer.

It is beyond the scope of the present paper to revisit and significantly expand this dynamical/modelling discussion: the project from which the climate model simulations are taken finished about five years ago. The opportunity in the present work is simply to re-use these GCM/RCM simulations – acknowledging their well-documented behaviours and limitations – to compare against a new regional synthesis of palaeo-observations (the paper should therefore be seen as 'paleo-data led' rather than 'modelling led' in terms of the conclusions it reaches). We believe this to be a reasonable approach to take as, in the absence of the resources to conduct new climate model experiments, the climate simulations used here remain the only published attempt at producing a high-resolution regional simulation of the Mediterranean with time-slices across the whole Holocene period. Insofar as the impact of specific local climate features is important (e.g., complex topography and coastlines), they remain the only dataset available for doing this level of detailed model-data inter-comparison in the region. We also note that the Brayshaw et al. (2010) paper compares the GCM results to other modelling work (PMIP) and palaeo-climate reconstructions available at the time (e.g., Brewer, Rimbu, etc) and, on balance of evidence, cautiously suggests an NAO-negative like state in the mid-Holocene (we would actually prefer to refer to a southerly shift in the North Atlantic storm track rather than the NAO). This stands in contrast to the more recent Mauri et al. publication (which makes no reference to these earlier publications).

We therefore seek to take on board the reviewer's concerns about the framing of the paper and its contextualization principally by improving the text in Section 2: • Making it clearer that this is a 're-use' of an existing model dataset; • Explicitly stating that the ocean dynamics are assumed to be invariant over time (strictly we 'fix' the oceanic fluxes of heat, as already noted in the paper, though we recognize that the implications of this may not be immediately recognized by all readers); • Refer more explicitly to the detailed analysis provided in previous work on this dataset (e.g. for changes in atmospheric circulation and comparison/justification of the modelling approach, e.g. compared to PMIP); • Emphasises the nature of the 'added value' of regional downscaling (i.e. the resolution of local impacts such as complex topography).

3) Conclusions: The conclusions could be presented in a more clear way. The last paragraph in the conclusions looks also quite convoluted, and some of the conclusions are not really based on the results presented here. For instance, the authors conclude that the regional model represents better the atmospheric dynamics, and therefore precipitation. This can be somehow expected, but it has not been shown in this study, and in particular, it has not been shown that the particular model set-up used here (with a slab ocean) is indeed better. We agree with the reviewer that this paragraph was unclear and perhaps a 'too general' conclusion to be drawn from the evidence presented. We have therefore clarified the text. The key issue we wish to highlight is that the RCM output provides a better representation of local/regional processes. Notwithstanding the difficulties of correctly modeling large-scale climate change over the Holocene (with GCMs), we believe that regional downscaling may still be valuable in facilitating model-data comparison in regions/locations known to be strongly influenced by local effects (e.g., complex topography).

4) The title is a bit misleading, as the study is basically about precipitation changes and not 'climate' in general. OK, corrected as follows: Precipitation changes in the Mediterranean basin during the Holocene from terrestrial and marine pollen records: a model/data comparison

5) Abstract has many elements of introduction, including recommendations, like the use of transient simulations, which turn out to be correct but that they are not really substantiated by the results described in this manuscript. My criticisms is to some extent a matter of taste, but I think that the abstract should be succinct and mainly describing the methods, results and conclusions. Introductory remarks should go in the introduction, and final speculations or recommendations, in the main text OK, corrected.

6) Lines 125-132 This is a repetition of a previous paragraph on the same page OK, corrected.

7) It may be interesting to know the time resolution of the proxy records Yes, I have added the time resolution in table 1 for the two periods selected and the entire sequence.

8) line 239 I think that the reference to Mauri et al (2015) is not correct. The reference Mauri, et al. (2015) is correct.

9) line 242 Mauri et al used a reconstruction method based on plant functional types. Should the reader expect differences to the reconstruction method used here ? Could some of the differences to the present results due to the different methodology ? Mauri et al. use the MAT with the plant functional type scores instead of the pollen assemblages; we use the MAT with the pollen assemblages; so yes, it can produce different results because different methods can produce different results (Brewer et al., 2008, Peyron et al., 2013).

10) line 266 'Mediterranean, and dry conditions above 45_N during the early Holocene, while the opposite' North of 45N I do not understand what you mean exactly; therefore I have corrected the sentence as follows: Our reconstructions are in agreement with Mauri et al. (2015), with dry summer conditions above 45°N during the early Holocene and wet summer conditions over much of the south-central Mediterranean south of 45°N.

11) Caption Figure 3. which is the reference period to calculate the simulated precipitation anomalies? Anomalies are taken with respect to present-day control run. Caption updated.

Second reviewer 1) I feel that there is a missed opportunity in using the model output to understand the atmospheric drivers of the changes in spatial pattern. As noted in

the response to reviewers, it is beyond the scope of the present paper to discuss the atmospheric drivers at length beyond that presented in Brayshaw et al. 2010, 2011a, and 2011b. I would like to see the goals more clearly stated, and clearly referred to throughout the paper. See reviewer 1, point 1. The primary novelty in this work is the new paleo-observations synthesis and its comparison – at regional/local level – with the climate model data. The text has, however, been clarified to direct interested readers to those works.

2) I would like to see more discussion about the choices of spatial pattern and of time period; For the time period, I don't really understand why the authors did not look across the entire Holocene, but instead focused on two, quite long time periods. Are these gradients only a feature of the time periods chosen? What was the variation outside (or even within) these periods? Given that one of the papers they cite has already completed full Holocene reconstructions (Mauri et al., 2015), and that there is interest in full Holocene/Glacial transient GCM simulations, this snapshot approach appears to be somewhat limited. At the very least, it would be good to have a better justification for the choices than "to aid interpretability I have also looked at it in a continuous way for the Holocene (not provided here), but it will be the topic of another paper. Here we focus on spatial patterns, and we have chosen these periods because they are different enough to be simulated by the regional model (which is not transient). From a climate-modelling perspective, the rationale for the grouping of the time-slices is a practical one (as noted above, we are unable to perform additional experiments to extend the dataset at this time). As outlined at ∼line243, the change in 'forcing' between adjacent time-slices is small and, as such, changes are difficult to detect robustly given the data available. Grouping the time-slices together into 'mid-Holocene' and 'late-Holocene' experiments therefore makes best use of the data available. The text in Section 2 (model description, ∼line 243) has been modified to emphasise the rationale for this decision. The text has also been changed as follows: This study aims to reconstruct and evaluate N-S and W-E climate conditions for the Mediterranean basin, over two key periods in the Holocene, 8000-6000 cal yrs BP, corresponding to the "Holocene climate optimum" and 4000-2000 cal yrs BP corresponding to a trend toward more dry conditions.

3) The choice of precipitation as a variable for comparison also needs better justification. The authors state (line 416) that using precipitation instead of moisture indices may be why there is a model/data mismatch, and some form of moisture index has been proposed as a better quantity for pollen reconstructions elsewhere (Bartlein et al., 2011). Please note that the Bartlein et al. (2011) paper is a synthesis at a world scale of "old" pollen inferred climate reconstructions done for different regions (Europe...). No new reconstruction has been done in the Bartlein et al. (2011) paper. Given this, and that alpha is routinely reconstructed from pollen, why not use this instead?. We use precipitation because our aim was to compare with the model outputs. In the regional model, the reconstruction of the moisture index was not available. And if not, please justify the use of precipitation, given its limitation as a reconstructed variable The use of precipitation parameters (annual and seasonal) seems robust for the Mediterranean area (Mauri et al., 2015; Peyron et al., 2011, 2013, Magny et al., 2013) and precipitation reconstructions are particularly important for the Mediterranean region given that precipitation rather than temperature represents the dominant controlling factor on Mediterranean environmental system during the early to mid-Holocene (Renssen et al., 2012). Text has been modified as follows: "precipitation reconstructions are particularly important for the Mediterranean region given that precipitation rather than temperature represents the dominant controlling factor on Mediterranean environmental system during the early to mid-Holocene (Renssen et al., 2012)". The use of precipitation parameters (annual and seasonal) seems robust for the Mediterranean area (Mauri et al., 2015; Peyron et al., 2011, 2013, Magny et al., 2013)."

4) Line 28. The abstract could be shortened and made more concise – there is some repetition (e.g. lines 39-40 and lines 60-61) OK, corrected, cf reviewer1.

5) Line 34 (and elsewhere). Is the pattern a gradient or dipole? These are not to my understanding the same thing, as one represents a trend, and the other represents a pattern of two opposing centers. Please either use one or the other, or state more clearly which is being referred to at any time. OK, checked and corrected.

6) Line 38-40. What is the aim of the comparison? Changed as follows: "For the same time intervals, site-based pollen-inferred precipitation estimates were compared with an existing database from a regional-scale downscaling of a set of global climate-model simulations. The high-resolution detail achieved through the downscaling is found to assist with comparing 'site-based' paleo-observations with gridded model data, and the climate model outputs and pollen-inferred precipitation estimates show remarkably good overall correspondence (although many simulated patterns are of marginal statistical significance)."

7) Lines 47-51. This section needs some rewriting to make it clear when the authors are referring to conditions being drier in one region than another, or that the anomalies are drier compared to another time period. Corrected as: "During the early Holocene, relatively wet conditions occurred in the south-central and eastern Mediterranean region, while drier conditions prevailed from 45°N northwards. Then these patterns appear to reverse during the late Holocene, with similar to present day or slightly drier than present day conditions in the south-central, but more sites from the northern part of the Mediterranean basin are needed to further substantiate these observations."

8) Line 61. In what sense is HadSM3 dynamic? (and what is HadSM3 as opposed to the other models shown here) This was an error. It has now been removed.

9) Lines 90-92. Needs a citation Magny et al. (2013) has been added.

10) Line 93. Which sites in N. Italy? Citation, please. Peyron et al. (2011, 2013) has been added.

11) Lines 126-127. Why these periods? Why are they 'key'? Why not do this in a continuous way? Cf point 2, reviewer2.

12) Lines 133-134. "To critically assess the potential of the model setup: : :" This is a little fuzzy, but I assume that the goal is to discuss the regional climate model output, and the model parameters. However, I don't really feel that this was addressed in the discussion. There is some discussion of findings in other papers (e.g. Bosmans et al) but nothing about the setup used here. We agree that the text was unclear at this point. The limit to the scope of this paper is such that our main concern here is to compare the climate simulated by the models to that reconstructed from the observations. The text is therefore changed to: "... critically assess the consistency of the climate reconstructions revealed by these two complimentary routes."

13) Line 169. Arguably, pine is overrepresented in all sites. Why only exclude it for the marine sites? How big an impact does this have? The pollen signal recorded in marine cores reflects the regional vegetation across an area of several hundred square kilometers and pine pollen is particularly overrepresented (Heusser and Balsam, 1977; Dupont and Wyputta, 2003; Hooghiemstra et al., 1992, 2006). The reliability of the quantitative climate reconstruction from marine pollen spectra (with and without Pinus) has been tested using marine core-top samples from the Mediterranean in Combourieu-Nebout et al., 2009. Results shows that an adequate consistency between the present day observed and MAT estimations is shown for Psum and Pann values. In terrestrial pollen records, the signal is more local (depending of the size of the lake). Pinus is of course also overrepresented, but excluding it from the terrestrial assemblages doesn't make sense for the Holocene because pine can grow close to each site. We can exclude Pinus during glacial times, where we are sure it was exclusively long-distance transport.

Text has been modified as: The reliability of quantitative climate reconstructions from marine pollen records has been tested using marine core-top samples from the Mediterranean in Combourieu-Nebout et al. (2009), which shows an adequate consistency between the present day observed and MAT estimations for Pann and Psum values.

14) Line 187. I think I get what the authors are saying here, but given the increasing interest in transient simulations (e.g. Liu et al) and reconstructions (e.g. Marcotte et al), I'd like to see this choice justified a little better As noted above, the modelling work is taking advantage of an existing model-output database (to our knowledge the only attempt that has been made thus far to simulate the regional climate of the Mediterranean across the whole period). This has now been clarified and readers are directed to appropriate previous publications for further discussion of the modelling framework.

15) Line 190. The model uses the pre-industrial period as a baseline for anomalies whereas I assume the pollen reconstructions use the late 20th century, although this is not specified. It includes both long-term averages (1961-90) and time series for rainfall. How much will this affect the offset between model and data. How big are the reconstructed anomalies relative to any change between these periods? We agree that it's an important point; changes in climate are now expressed as differences with respect to the present day control run. Text has been changed as follows: "In contrast to existing model simulations, changes in climate are expressed here as differences with respect to the present day (1960-1990) and not with respect to pre-industrial. We suggest it may be better to use 'present day' to be in closer agreement with the pollen data (modern samples) which use the late 20th century long-term averages (1961-1990). However, there are some quite substantial differences between model runs under 'present day' and 'preindustrial' forcings (Fig. 4)."

16) Line 206. The results are more single points in time, rather than climate trends Changed 'trends' to 'patterns'.

17) Lines 217-219. This seems like it would be more appropriate in the introduction OK, corrected.

18) Line 231. What scaling issues? Wu et al. undertook a reconstruction at a world scale: it's difficult to distinguish in their figures what happened exactly in the central Mediterranean to depict a possible north-south pattern.

19) Line 247. It is difficult to see visually how there is good corroboration between these results and Mauri et al. Would it be possible to carry out a one-to-one comparison of values, and test the differences? To carry out a one-to-one comparison of values, we need to have access to Mauri's data, which is not the case, therefore it was not possible to test the differences (furthermore this was not the topic/focus of this paper). Further – as both Mauri et al, and this study present statistical climate reconstructions from pollen data, it is hard to see how the agreement between them supports the robustness of the results. In contrast to Mauri et al., our study also performed climate reconstruction from marine pollen cores. Because the scale of our figures and those of Mauri et al. were different, we finally decided to remove the Mauri et al. reconstruction in the figure 2.

20) Line 351. How can you tell that the data-model agreement is good? Again, some point-by-point comparison would help (and help highlight where the main differences are) It is only visual. We agree that it is not the top, however we did not have time to produce metrics to compare simulations and reconstructions.

21) Line 435. How would the snowpack affect different methods? I can see that it might affect proxies differently, but not methods. Yes, we agree and it is corrected.

22) Line 486. It is hard to disagree with a call for higher resolution in climate models, but how exactly will this help? What processes will be better represented, allowing for better climate simulation? This text has been amended to better reflect the scope and content of the paper. In particular, while the authors do believe that high-resolution global models are likely to be part of the solution (e.g. improved representation of processes such as blocking, storms, teleconnections etc), this is not directly the subject of the paper. The revised discussion therefore now focuses on the potential role for regional high-resolution models in providing a better representation of complex terrain to reconcile site-specific model vs palaeo-obs discrepancies.

Please also note the supplement to this comment:
http://www.clim-past-discuss.net/cp-2016-65/cp-2016-65-AC1-supplement.pdf

[Figure]

[Figure]

**Supplement:**

[revised manuscript text omitted]

---

## Editor Comment (EC2) · J. Guiot (Editor) · 21 Nov 2016

Dear authors Thank you for your long reply. But it is very difficult to follow. The comments of the authors are mixed with your replies and I had to compare your letter to the original reviews to separate comments and replies. It is very annoying. Moreover your letter is not structured and refers to unexisting numbers in the reviews. So, you start with a reply to rev#2 then you go to rev#1 and go back to rev#2. Point 15 of rev#2 does not exist. Finally (BUT VERY IMPORTANT), as I mentioned in my previous message, I NEED a "track change mode" revised version of the ms linked to your reply. This will greatly accelerate the review process.

By the way, I am not convinced by some of your replies. For example:

[Figure]

When rev#1 asks you to identify new conclusions according to previous studies, he means according to results on the climate changes and not only according to the fact that you use two periods and a RCM. By the way, Roberts et al (2011) compared model simulations and data on four periods separated by 2000 years.

Your reply to rev#2 concerning the use of precipitation instead of moisture index is not satisfying. The fact that the data used by Bartlein are "old" data is not an argument. The argument that alpha is not available from RCM is also not correct (particularly for a RCM). Mediterranean vegetation is not limited by precipitation but by soil water. I agree that monsoon is a matter of precipitation, but when you compare data and models you must do it on common signals. Then please rework your arguments.

I agree also with the comment of rev#1 that your message must be clearer. Finally you have the opportunity to have data and RCM simulations, you have to use this chance and to deepen your discussion with more mechanistic interpretation.

Best regards Joel Guiot

---

## Author Comment (AC2) · 28 Nov 2016

Paper: The climate of the Mediterranean basin during the Holocene from terrestrial and marine pollen records: A model/data comparison

By Odile Peyron et al, Clim. Past Discuss: cp-2016-65

**First reviewer**

1)      It is not easy to identify what are the new conclusions, or what new information the data or the analysis is providing beyond of what is already new.
**To be more precise, some sentences have been added in the introduction to clarify the goals: "The first originality of our approach is that we estimate the magnitude of precipitation changes and reconstruct climatic trends across the Mediterranean using both terrestrial and marine high-resolution pollen records. The signal reconstructed is then more regional than in the studies based on terrestrial records alone. Moreover, this study aims to reconstruct precipitations patterns for the Mediterranean basin over two key periods in the Holocene, while the existing large-scale quantitative paleoclimate reconstructions for the Holocene are often limited to the mid-Holocene - 6000 yrs BP- (Cheddadi et al., 1997; Bartlein et al., 2011; Mauri et al., 2014), except the climate reconstruction for Europe proposed by the study of Mauri et al. (2015). The second originality of our approach is that we propose a data/model comparison based on: (1) two time-slices and not only the mid-Holocene, a standard benchmark time period for this kind of data–model comparison; (2) a high resolution regional model (RCM) which provides a better representation of local/regional processes and helps to better simulate the localized, "patchy", impacts of Holocene climate change, when compared to coarser global GCMs (e.g. Mauri et al., 2014); (3) changes in seasonality, particularly changes in summer atmospheric circulation which have not been widely investigated (Brayshaw et al., 2011)."**

**Some sentences have also been added in the abstract to clarify what is new in terms of results: With regard to the existence of a west-east precipitation dipole during the Holocene, our pollen-based climate data show that the strength of this dipole is strongly linked to the seasonal parameter reconstructed; early Holocene summers show a clear east-west division, with summer precipitation having been highest in Greece and the eastern Mediterranean and lowest over the Italy and the western Mediterranean. Summer precipitation in the east remained above modern values, even during the late Holocene interval.**
**In contrast, winter precipitation signals are less spatially coherent during the early Holocene but low precipitation is evidenced during the early and late Holocene.**

2)      In the model set-up used by the authors there are some open questions. For instance, they use a slab ocean that ignores the ocean dynamics, but are the simulated sea-surface temperatures comparable to the temperatures simulated in global couple simulations for the mid-Holocene? what could be the role of the dynamics of the North Atlantic in determining the precipitation patterns in Europe? I am aware that a full coupled simulation over the Holocene could be out of the scope of the present study in terms of computer resources, but some type of validation or discussion of the possible shortcoming of the simulation set-up should be addressed. More importantly, I think, would be to identify which aspects of the regional modelling provide an added value relative to the global model results presented in of Mauri et al. . The manuscript includes just a comment in passing about the heterogeneity of simulated precipitation changes in the Balkans, but

this is not really followed trough. For instance, one of the mechanisms that may explain the pattern or precipitation changes are shifts in the North Atlantic storm tracks. Is the regional model able to represent the storm tracks more realistically than the global models ? Is the representation of present-day precipitation better in the regional model than in the ensemble of CMIP5 global models ? I would assume the answer is yes, but it it would be nice to see it discussed in the manuscript as well. On the other hand, the slab ocean is likely not able to realistically represent the meridional sea-surface temperatures in the Atlantic.This may affect the intensity and extent of the African Monsoon and its changes over the Holocene. Could this limitation influence the simulation of t summer precipitation changes in the Mediterranean?

**This comment raises several important issues, which we attempt to disentangle as follows.**

**We agree with the reviewer that there are indeed limitations in the climate modeling approach used here. These were discussed at length in previous publications, as cited from the present article. Brayshaw et al., 2010 (Phil Trans A), in particular, goes into detail on (A) evaluating the relative merits and difficulties of the modeling approach compared to others (such as PMIP), including the role of embedded high-resolution regional modelling and (B) discussing the physical atmospheric drivers of winter-time change such as storm tracks, Hadley cell expansion/contraction and teleconnections from the Indian Ocean. Brayshaw et al., 2011 (Holocene) provides a wider review (both summer and winter), discussing of the impact of the GCM's simulation of tropical Atlantic SSTs (on, e.g., the summer expansion of the African monsoon) as suggested by the reviewer.**

**It is beyond the scope of the present paper to revisit and significantly expand this dynamical/modelling discussion: the project from which the climate model simulations are taken finished about five years ago. The opportunity in the present work is simply to re-use these GCM/RCM simulations – acknowledging their well-documented behaviours and limitations – to compare against a new regional synthesis of palaeo-observations (the paper should therefore be seen as 'paleo-data led' rather than 'modelling led' in terms of the conclusions it reaches). We believe this to be a reasonable approach to take as, in the absence of the resources to conduct new climate model experiments, the climate simulations used here remain the only published attempt at producing a high-resolution regional simulation of the Mediterranean with time-slices across the whole Holocene period. Insofar as the impact of specific local climate features is important (e.g., complex topography and coastlines), they remain the only dataset available for doing this level of detailed model-data inter-comparison in the region.**

**We also note that the Brayshaw et al. (2010) paper compares the GCM results to other modelling work (PMIP) and palaeo-climate reconstructions available at the time (e.g., Brewer, Rimbu, etc) and, on balance of evidence, cautiously suggests an NAO-negative like state in the mid-Holocene (we would actually prefer to refer to a southerly shift in the North Atlantic storm track rather than the NAO). This stands in contrast to the more recent Mauri et al. publication (which makes no reference to these earlier publications).**

**We therefore seek to take on board the reviewer's concerns about the framing of the paper and its contextualization principally by improving the text:**
- **Making it clearer that this is a 're-use' of an existing model dataset;**
- **Explicitly stating that the ocean dynamics are assumed to be invariant over time (strictly we 'fix' the oceanic fluxes of heat, as already noted in the paper, though we recognize that the implications of this may not be immediately recognized by all readers);**

- **Refer more explicitly to the detailed analysis provided in previous work on this dataset (e.g. for changes in atmospheric circulation and comparison/justification of the modelling approach, e.g. compared to PMIP);**
- **Emphasises the nature of the 'added value' of regional downscaling (i.e. the resolution of local impacts such as complex topography).**

3)    Conclusions: The conclusions could be presented in a more clear way. The last paragraph in the conclusions looks also quite convoluted, and some of the conclusions are not really based on the results presented here. For instance, the authors conclude that the regional model represents better the atmospheric dynamics, and therefore precipitation. This can be somehow expected, but it has not been shown in this study, and in particular, it has not been shown that the particular model set-up used here is indeed better.
**We agree with the reviewer that this paragraph was unclear and perhaps a 'too general' conclusion to be drawn from the evidence presented. We have therefore clarified the text.**
**The key issue we wish to highlight is that the RCM output provides a better representation of local/regional processes. Notwithstanding the difficulties of correctly modeling large-scale climate change over the Holocene (with GCMs), we believe that regional downscaling may still be valuable in facilitating model-data comparison in regions/locations known to be strongly influenced by local effects (e.g., complex topography).**

4)    The title is a bit misleading, as the study is basically about precipitation changes and not 'climate' in general.
**OK, corrected as follows: Precipitation changes in the Mediterranean basin during the Holocene from terrestrial and marine pollen records: a model/data comparison**

5)    Abstract has many elements of introduction, including recommendations, like the use of transient simulations, which turn out to be correct but that they are not really substantiated by the results described in this manuscript. My criticisms is to some extent a matter of taste, but I think that the abstract should be succinct and mainly describing the methods, results and conclusions. Introductory remarks should go in the introduction, and final speculations or recommendations, in the main text
**OK, corrected.**

6)    Lines 125-132 This is a repetition of a previous paragraph on the same page
**OK, corrected.**

7)    It may be interesting to know the time resolution of the proxy records
**Yes, I have added the time resolution in table 1 for the two periods selected and the entire sequence.**

8)    line 239 I think that the reference to Mauri et al (2015) is not correct.
**The reference Mauri, et al. (2015) is correct.**

9)    line 242 Mauri et al used a reconstruction method based on plant functional types. Should the reader expect differences to the reconstruction method used here ? Could some of the differences to the present results due to the different methodology ?
**Mauri et al. use the MAT with the plant functional type scores instead of the pollen assemblages; we use the MAT with the pollen assemblages; so yes, it can produce different results because different methods can produce different results (Brewer et al., 2008, Peyron et al., 2013).**

10) line 266 'Mediterranean, and dry conditions above 45_N during the early Holocene, while the opposite' North of 45N
**I do not understand what you mean exactly; therefore I have corrected the sentence as follows: Our reconstructions are in agreement with Mauri et al. (2015), with dry summer conditions above 45°N during the early Holocene and wet summer conditions over much of the south-central Mediterranean south of 45°N.**

11) Caption Figure 3. which is the reference period to calculate the simulated precipitation anomalies?
**Anomalies are taken with respect to present-day control run. Caption updated.**

Paper: The climate of the Mediterranean basin during the Holocene from terrestrial and marine pollen records: A model/data comparison

By Odile Peyron et al, Clim. Past Discuss: cp-2016-65

**Second reviewer**

**An important point came out in Review 2 (point 15), concerning the use of the preindustrial baseline. In the model simulations, we have always used PREIND as the baseline because the climate forcing before then, over the Holocene, is mostly orbital; in contrast to the industrial period where it is mostly-greenhouse gas. It does, however, have some impact on the precipitation signals we are discussing here (new figure 4).**

**Therefore, in this revised version, we have changed our model-data synthesis (Fig. 3) and have taken present day in the control run instead of preindustrial to be in better agreement with the pollen data (the pollen data precipitation is best seen as 'anomalies relative to present day 1960-1990').**

**This changed our results, particularly the winter precipitation output which suggest now dry conditions in the Early Holocene compared to the previous version.**

1)      I feel that there is a missed opportunity in using the model output to understand the atmospheric drivers of the changes in spatial pattern.
     **As noted in the response to reviewer 1, it is beyond the scope of the present paper to discuss the atmospheric drivers at length beyond that presented in Brayshaw et al. 2010, 2011a, and 2011b.**
     I would like to see the goals more clearly stated, and clearly referred to throughout the paper.
     **We did it in the abstract, introduction and conclusion: see the reply to reviewer 1, point 1. The primary novelty in this work is the new paleo-observations synthesis and its comparison – at regional/local level – with the climate model data. The text has, however, been clarified to direct interested readers to those works.**

2)      I would like to see more discussion about the choices of spatial pattern and of time period; For the time period, I don't really understand why the authors did not look across the entire Holocene, but instead focused on two, quite long time periods. Are these gradients only a feature of the time periods chosen? What was the variation outside (or even within) these periods? Given that one of the papers they cite has already completed full Holocene reconstructions (Mauri et al., 2015), and that there is interest in full Holocene/Glacial transient GCM simulations, this snapshot approach appears to be somewhat limited. At the very least, it would be good to have a better justification for the choices than "to aid interpretability
     **I have also looked at it in a continuous way for the Holocene. The results are not provided here because it will be the topic of another paper.**
     **Here we focus on spatial patterns, and we have chosen these periods because they are different enough to be simulated by the regional model (which is not transient). From a climate-modelling perspective, the rationale for the grouping of the time-slices is a practical one (as noted above, we are unable to perform additional experiments to extend the dataset at this time). As outlined, the change in 'forcing' between adjacent time-slices is small and, as such, changes are difficult to detect**

robustly given the data available. Grouping the time-slices together into 'mid-Holocene' and 'late-Holocene' experiments therefore makes best use of the data available. The text in Section 2 (model description, ~line 243) has been modified to emphasize the rationale for this decision.

**The text has also been changed as follows: This study aims to reconstruct and evaluate N-S and W-E climate conditions for the Mediterranean basin, over two key periods in the Holocene, 8000-6000 cal yrs BP, corresponding to the "Holocene climate optimum" and 4000-2000 cal yrs BP corresponding to a trend toward more dry conditions.**

3) The choice of precipitation as a variable for comparison also needs better justification. The authors state (line 416) that using precipitation instead of moisture indices may be why there is a model/data mismatch, and some form of moisture index has been proposed as a better quantity for pollen reconstructions elsewhere (Bartlein et al., 2011).

**Please note that the Bartlein et al. (2011) paper is a synthesis at a world scale of "old" pollen inferred climate reconstructions done for different regions (Europe…). No new reconstruction has been done in the Bartlein et al. (2011) paper. Sorry to insist, but these old results are still used in a lot of recent model-data comparison to check model outputs (eg Harrison et al., 2014), and but my feeling is that more work are needed to do more in depth including new data/proxies/methods.**

Given this, and that alpha is routinely reconstructed from pollen, why not use this instead? **We have calculated the moisture index from pollen data but we made the choice to use precipitation instead of alpha because our aim was to compare with the model outputs. Most often, GCM-data comparison are based on annual precipitation (Braconnot et al., 2012, Mauri et al., 2014, Harrison et al., 2015) and not on alpha (Harrison et al., 2014). Here too, the reconstruction of the moisture index with the RCM was unfortunately not available; it will not be available for this study because we don't have financial resources to conduct new climate model experiments. We agree with the reviewer that it's an important point to test in future experiments.**

And if not, please justify the use of precipitation, given its limitation as a reconstructed variable.

**The use of precipitation parameters (annual and seasonal) seems robust for the Mediterranean area (Mauri et al., 2015; Peyron et al., 2011, 2013, Magny et al., 2013); precipitation reconstructions are particularly important for the Mediterranean region given that precipitation rather than temperature represents the dominant controlling factor on Mediterranean environmental system during the early to mid-Holocene (Renssen et al., 2012).**

**Text has been modified as follows: "precipitation reconstructions are particularly important for the Mediterranean region given that precipitation rather than temperature represents the dominant controlling factor on Mediterranean environmental system during the early to mid-Holocene (Renssen et al., 2012)". The use of precipitation parameters (annual and seasonal) seems robust for the Mediterranean area (Mauri et al., 2015; Peyron et al., 2011, 2013, Magny et al., 2013)."**

4) Line 28. The abstract could be shortened and made more concise – there is some repetition (e.g. lines 39-40 and lines 60-61)
**OK, corrected, cf reviewer1.**

5) Line 34 (and elsewhere). Is the pattern a gradient or dipole? These are not to my understanding the same thing, as one represents a trend, and the other represents a pattern of two opposing centers. Please either use one or the other, or state more clearly which is being referred to at any time.
**OK, checked and corrected.**

6)    Line 38-40. What is the aim of the comparison?
**Changed as follows: "For the same time intervals, site-based pollen-inferred precipitation estimates were compared with an existing database from a regional-scale downscaling of a set of global climate-model simulations. The high-resolution detail achieved through the downscaling is found to assist with comparing 'site-based' paleo-observations with gridded model data, and the climate model outputs and pollen-inferred precipitation estimates show remarkably good overall correspondence (although many simulated patterns are of marginal statistical significance)."**

7)    Lines 47-51. This section needs some rewriting to make it clear when the authors are referring to conditions being drier in one region than another, or that the anomalies are drier compared to another time period.
**Corrected as: "During the early Holocene, relatively wet conditions occurred in the south-central and eastern Mediterranean region, while drier conditions prevailed from 45°N northwards. Then these patterns appear to reverse during the late Holocene, with similar to present day or slightly drier than present day conditions in the south-central, but more sites from the northern part of the Mediterranean basin are needed to further substantiate these observations."**

8)    Line 61. In what sense is HadSM3 dynamic? (and what is HadSM3 as opposed to the other models shown here)
**This was an error. It has now been removed.**

9)    Lines 90-92. Needs a citation
**Magny et al. (2013) has been added.**

10)   Line 93. Which sites in N. Italy? Citation, please.
**Peyron et al. (2011, 2013) has been added.**

11)   Lines 126-127. Why these periods? Why are they 'key'? Why not do this in a continuous way?
**Cf point 2, reviewer2.**

12)   Lines 133-134. "To critically assess the potential of the model setup: : :" This is a little fuzzy, but I assume that the goal is to discuss the regional climate model output, and the model parameters. However, I don't really feel that this was addressed in the discussion. There is some discussion of findings in other papers (e.g. Bosmans et al) but nothing about the setup used here.
**We agree that the text was unclear at this point. The limit to the scope of this paper is such that our main concern here is to compare the climate simulated by the models to that reconstructed from the observations. The text is therefore changed to: "... critically assess the consistency of the climate reconstructions revealed by these two complimentary routes."**

13)   Line 169. Arguably, pine is overrepresented in all sites. Why only exclude it for the marine sites? How big an impact does this have?
**The pollen signal recorded in marine cores reflects the regional vegetation across an area of several hundred square kilometers and pine pollen is particularly overrepresented (Heusser and Balsam, 1977; Dupont and Wyputta, 2003; Hooghiemstra et al., 1992, 2006). The reliability of the quantitative climate reconstruction from marine pollen spectra (with and without *Pinus*) has been**

**tested using marine core-top samples from the Mediterranean in Combourieu-Nebout et al., 2009. Results shows that an adequate consistency between the present day observed and MAT estimations is shown for Psum and Pann values.
In terrestrial pollen records, the signal is more local (depending of the size of the lake).** *Pinus* **is of course also overrepresented, but excluding it from the terrestrial assemblages doesn't make sense for the Holocene because pine can grow close to each site. We can exclude** *Pinus* **during glacial times, where we are sure it was exclusively long-distance transport.**

Text has been modified as: **The reliability of quantitative climate reconstructions from marine pollen records has been tested using marine core-top samples from the Mediterranean in Combourieu-Nebout et al. (2009), which shows an adequate consistency between the present day observed and MAT estimations for Pann and Psum values.**

14) Line 187. I think I get what the authors are saying here, but given the increasing interest in transient simulations (e.g. Liu et al) and reconstructions (e.g. Marcotte et al), I'd like to see this choice justified a little better
**As noted above, the modelling work is taking advantage of an existing model-output database (to our knowledge the only attempt that has been made thus far to simulate the regional climate of the Mediterranean across the whole period). This has now been clarified and readers are directed to appropriate previous publications for further discussion of the modelling framework.**

15) Line 190. The model uses the pre-industrial period as a baseline for anomalies whereas I assume the pollen reconstructions use the late 20th century, although this is not specified. **It includes both long-term averages (1961-90) and time series for rainfall.** How much will this affect the offset between model and data. How big are the reconstructed anomalies relative to any change between these periods?
**We agree that it's an important point; changes in climate are now expressed as differences with respect to the present day control run.**
**Text has been changed as follows: "In contrast to existing model simulations, changes in climate are expressed here as differences with respect to the present day (1960-1990) and not with respect to pre-industrial. We suggest it may be better to use 'present day' to be in closer agreement with the pollen data (modern samples) which use the late 20th century long-term averages (1961-1990). However, there are some quite substantial differences between model runs under 'present day' and 'preindustrial' forcings (Fig. 4)."**

16) Line 206. The results are more single points in time, rather than climate trends
**Changed 'trends' to 'patterns'.**

17) Lines 217-219. This seems like it would be more appropriate in the introduction
**OK, corrected.**

18) Line 231. What scaling issues?
**Wu et al. undertook a reconstruction at a world scale: it's difficult to distinguish in their figures what happened exactly in the central Mediterranean to depict a possible north-south pattern.**

19) Line 247. It is difficult to see visually how there is good corroboration between these results and Mauri et al. Would it be possible to carry out a one-to-one comparison of values, and test the differences? **To carry out a one-to-one comparison of values, we**

**need to have access to Mauri's data, which is not the case, therefore it was not possible to test the differences (furthermore this was not the topic/focus of this paper).** Further – as both Mauri et al, and this study present statistical climate reconstructions from pollen data, it is hard to see how the agreement between them supports the robustness of the results.

**In contrast to Mauri et al., our study also performed climate reconstruction from marine pollen cores. Because the scale of our figures and those of Mauri et al. were different, we finally decided to remove the Mauri et al. reconstruction in the figure 2.**

20) Line 351. How can you tell that the data-model agreement is good? Again, some point-by-point comparison would help (and help highlight where the main differences are)
**It is only visual. We agree that it is not the top, however we did not have time to produce metrics to compare simulations and reconstructions.**

21) Line 435. How would the snowpack affect different methods? I can see that it might affect proxies differently, but not methods.
**Yes, we agree and it is corrected.**

22) Line 486. It is hard to disagree with a call for higher resolution in climate models, but how exactly will this help? What processes will be better represented, allowing for better climate simulation?
**This text has been amended to better reflect the scope and content of the paper. In particular, while the authors do believe that high-resolution global models are likely to be part of the solution (e.g. improved representation of processes such as blocking, storms, teleconnections etc), this is not directly the subject of the paper. The revised discussion therefore now focuses on the potential role for regional high-resolution models in providing a better representation of complex terrain to reconcile site-specific model vs palaeo-obs discrepancies.**

---

## Author Response (AR2)

Paper: The climate of the Mediterranean basin during the Holocene from terrestrial and marine pollen records: A model/data comparison

By Odile Peyron et al, Clim. Past Discuss: cp-2016-65

**Reply to the reports**

**Report1**

I would suggest to pay attention to three points that would strike the reader.

The first point is the abstract. It is too long and some sentences /paragraph are repetitive. I think it should be tightened and shortened, maybe by 1/3 or so. It is written very 'generoulsy' and certainly the sentences can be shortened and the whole abstract written more to the point.

**We have shortened the abstract and removed the repetitive sentences.**

The second point, in the introduction, is related to the discussions of previous proxy studies. The introduction sets off by explaining all what is known about the evolution of precipitation in Europe over the Holocene, which reads to be quite a lot (early Holocene, Midholocene, north-south, east-west dipoles), but then, a bit surprisingly, when highlighting the innovations of this study the authors point out that most previous studies are focused on the Midholocene (lines 147). The reader will wonder how we could know so much if most studied are limited to the Mid-Holocene.

One innovation of this paper is to provide quantitative estimates of precipitation, and it's true that previous large-scale quantitative paleoclimate reconstructions are limited to the mid-Holocene - 6000 yrs BP (except the papers by Mauri et al 2015, and by Guiot and Kaniewski, 2015). The other studies we mentioned in the introduction are based on different proxies from which quantitative estimates of precipitation are often not available or are not at large scale.

Regarding the modelling side, I think that the authors do not do a favor themselves. They present results from a regional simulation backed by the reasoning that a high-resolution model is necessary to better simulate precipitation, but on the other hand also discuss that the results of the regional model cannot deviate much from the driving global model, for instance considering the question of the extension of the African Monsoon in the Mid-Holocene. Again, the reader would wonder why is a regional model necessary in the first place, and why the authors could not look into global coupled simulations.

For any given climate model, there is a trade off in computational expense between resolution (number of grid boxes) and duration of run (number of years of simulation). Ideally, one needs both resolution and duration – resolution to represent fine scale features and processes (e.g., topography, complex coastlines, small scale dynamical processes) plus duration to robustly sample climate conditions (e.g., 'natural' chaotic variations on timescales of years or decades). As a rule of thumb, a minimum of a few decades are needed to provide any meaningful sample of climate variability (one could easily argue from recent literature that much much longer simulations are actually required), and one might expect climate models to reliably resolve spatial processes to some extent on the order of a few-to-several grid boxes in size.

At the time the simulations here were produced, a typical global model (GCM) capable of several hundred years of palaeoclimate simulations in a 'standard-sized' research project might have a grid-box resolution of a few hundred km (~200-300km for the model here, comparable to PMIP2). At this resolution we were able to produce a total of a few thousand 'useful' model years (not all of

which have been reported in the literature). We consider that these are capable of providing 'useful' spatial information at ~1000km ('useful' is in inverted commas because it there is no absolute guarantee that the simulation is accurate to reality). To halve the spatial scale of this 'useful' data would require an 8-fold increase in computational expense, which would have massively restricted the number of models years that could be completed in the computing resource available.

By using the regional model (~50km resolution but only covering a limited domain so less grid boxes than a global model at the same resolution), we were able to provide 'useful' spatial information down to, say, scales typically around ~200km. While this is still quite 'large', it is, we believe, still more useful than the raw global model data when comparing to palaeo-observation data which is often inherently local (i.e., depends on very specific local conditions perhaps down to a ~few km in scale). In this sense, it the regional model can add value, particularly in a complex region like the Mediterranean (e.g., complex coastlines, mountains etc).

The regional model, however, takes the large scale circulation (> few 1000 km) produced by the global model as an input assumption: it cannot adjust this as part of the regional simulation. In this case, if the global model does not simulate an extension of the West African Monsoon (a feature much larger than ~1000km), then it is hard for the regional model to do so either. In this sense, the regional model cannot offer additional value.

This is an issue that would be common to all one-way dynamical downscaling with regional climate models, so is not unique to this paper. We therefore do not think it appropriate or helpful to go into a general discussion of this in the paper, nor do we seek to provide precise guidance as to which spatial scales are accurately represented in either of the models used (the numbers given above are approximate guidelines and can be considered to be based on expert judgement rather than quantitative analysis - see, e.g., Cannon et al 2015 Renewable Energy and Cannon et al in press for MetZet for similar exercises but in a very different context). Furthermore, as already indicated in the present paper, a more in-depth, processed-based discussion of the circulation changes has already been provided in several papers and books (Brayshaw et al 2010, Phil Trans A; Brayshaw et al 2011, Holocene; Brayshaw et al 2011 WLC book) so are not discussed in detail here.

**Report2**

I thank the authors for their detailed reply to my original comments and for the changes that they have made to the manuscript. While the paper has improved, I have a few further comments that I would like to see addressed prior to publication:

Line 33 (and elsewhere): "regional/local level". Please define what is meant by these terms – I'm not convinced that a regional GCM can really inform local scale processes. See above discussion for detail. Line 33: "regional/local level" replaced with "regional (few ~100km) level".

Line 57: "general drying trend". This is not really a trend, as it is based on two points in time *It is not based on two points in time: all the values estimated inside the two time slices of 2000 years each have been have been averaged to produce the values in figures 2 and 3.*

Lines 171-172. This seems like a small number of sites, and checking against the EPD records,

suggest that there should be several more in both time periods. Clearly there has been some selection – please state what the criteria were.

We used the data acquired in the framework of our funded project (ANR), and we also have choosen sites for which multi-proxies and good age control were available.

Further, what are the time windows used to select samples? How much variation is there at sites within these windows?

The time window is the two time slices: all the values reconstructed available during these 2 time slices have been averaged to be compared with the model outputs. The variation within these windows depend of each site.

Text has been changes lines 235.

Lines 188-190: I don't really follow this justification for the MAT method. If you are using a non-robust statistic such as the mean, then I would think it is more susceptible to bias from higher noise. Please add some more detail here.

This method have been discussed in detail and compared with other methods in Peyron et al., 2011.

Line 197: What about the winter precipitation reconstructions? The MAT seems to overestimate the winter precipitation reconstructions by about 60mm in comparison with the observed values (Combourieu-Nebout et al., 2009). However, this study was based on 22 marine top cores; more samples are then needed to validate these results at the scale of the Mediterranean basin, particularly in the eastern part where only one marine top core was available. Text has been added.

Line 203: Did the authors merge the simulations for 2000 and 4000, and those for 6000 and 8000? Wouldn't it have been easier to choose 2000 and 8000 to maximize the differences? Please justify this choice.

The choice is motivated by two observations: 1) Long simulations are beneficial for robustly detecting differences in climate, and 2) the differences between adjacent timeperiods is small (both in terms of climate forcing and climate response). As such, it was decided that combining simulations together (40 model years per experiment) was a more robust method for sampling the qualitative change between middle and late Holocene rather than taking the end points (20 model years per experiment). As noted in the text, this follows the approach used in previously published work.

**Line 219/220 replaced:**

"These two experiments aid interpretability and increase the signal-to-noise ratio (the change in forcing between adjacent time-slices is relatively small, making it difficult to detect)."

With

"The combination of the simulations into two experiments (Mid- and Late- Holocene) rather than assessing the two extreme timeslices is intended to increase the signal-to-noise ratio by doubling the quantity of data in each experiment. This is necessary and possible as the change in forcing between adjacent time-slices is relatively small, making it difficult to detect differences between each individual simulations."

Line 212. What is HadSM3?

This is the name usually given to HadAM3 coupled to a slab ocean model. Line 200 replaced:

"... coupled to a slab ocean (Hewitt et al., ..." with "... coupled to a slab ocean (HadSM3; Hewitt et al., ..."

Line 228: What is being tested? The difference between simulations? The hatching representing statistical significance refers to the anomalies shown on the same plot – i.e., the difference between the experiment (either MidHolocene or LateHolocene) and the PresentDay control run – as per normal practice in the climate science literature. Additional text has been added both at line 400 and at the figure caption of Fig.3 to clarify.

Line 400 replaced "...compared to present values (in anomalies)..." with "...compared to the Present Day control run (in anomalies, with statistical significance hatched). "

Section 3 Results and Discussion. While the maps show some apparently convincing matches between the reconstructions and simulations, it is very difficult to judge these. It would really help to have a figure that shows the reconstructed and simulated values perhaps as a function of longitude. This could also include the model and reconstruction uncertainty, and would make it easier to follow the points made in the discussion, as well as the assertion of "a remarkable qualitative agreement"

We agree, but it will not be possible to build new figures: one of our author which did the model simulation is in sick leave for several months, so we don't have access to the simulated values to build a figure that shows the reconstructed and simulated values as a function of longitude or other.

Line 403. The author mention here (and elsewhere) that many of the changes are small and of marginal significance. However, even these null changes are of interest if found in both data and model. Again, a figure displaying the actual values would help. *Same as above*

Line 409. What is a level of significance of 70%. A p-value of 0.7? Yes. Line 409 replaced " level of significance of 70%" with "level of significance of 70% (p-value=0.7)".

Line 479. Data limitations. Thank you for including this section, which provides a great overview of some of the limitations. There are a couple of phrases that should be reviewed by an native English speaker.

These comments are surprising given that Belinda Gambin and Simon Goring, two native English speaker have reviewed all the text.

Changed as follows:

Line 493: replaced "it may be highlighting commenting on" with "it may be worth commenting on"

Line 502 replaced "All of these points may seem very picky on the ecology side, but they may have" with "Although these issues may initially appear to be of marginal importance, they may nevertheless have..."

Figures. Figures 2 and 3 carry much of the same data. Do the authors really need both? *Yes, we think that we need both to discuss more as clearly as we can the results.*

**Precipitation changes in the Mediterranean basin during**

[revised manuscript text omitted]

|--------------------------------------------------------------------------------------------------------------------------------------------|
Mediterranean to dry western Mediterranean                                      |
|                                                                                                                                            |

[revised manuscript text omitted]